# Global Convergence Guarantees for Federated Policy Gradient Methods with Adversaries

**Swetha Ganesh**                                                                        *swethaganesh@iisc.ac.in*
*Indian Institute of Science (IISc), Bengaluru 560012, India*
*Purdue University, West Lafayette, IN, 47907, USA*

**Jiayu Chen**                                                                          *jiayuc2@andrew.cmu.edu*
*Carnegie Mellon University (CMU), Pittsburgh, PA, 15289, USA*

**Gugan Thoppe**                                                                            *gthoppe@iisc.ac.in*
*Indian Institute of Science (IISc), Bengaluru 560012, India*

**Vaneet Aggarwal**                                                                         *vaneet@purdue.edu*
*Purdue University, West Lafayette, IN, 47907, USA*

**Reviewed on OpenReview:** *https://openreview.net/forum?id=EaOLrPORzM*

## Abstract

Federated Reinforcement Learning (FRL) allows multiple agents to collaboratively build a decision making policy without sharing raw trajectories. However, if a small fraction of these agents are adversarial, it can lead to catastrophic results. We propose a policy gradient based approach that is robust to adversarial agents which can send arbitrary values to the server. Under this setting, our results form the first global convergence guarantees with general parametrization. These results demonstrate resilience with adversaries, while achieving optimal sample complexity of order $\tilde{\mathcal{O}}\left(\frac{1}{N\epsilon^2}\left(1 + \frac{f^2}{N}\right)\right)$, where $N$ is the total number of agents and $f < N/2$ is the number of adversarial agents.

## 1 Introduction

Reinforcement Learning (RL) encompasses a category of challenges in which an agent iteratively selects actions and acquires rewards within an unfamiliar environment, all while striving to maximize the cumulative rewards. The Policy Gradient (PG) based approaches serve as an effective means of addressing RL problems, demonstrating its successful application in a diverse range of complex domains, such as game playing (Schrittwieser et al., 2020; Bonjour et al., 2022), transportation (Kiran et al., 2021; Al-Abbasi et al., 2019), robotics (Abeyruwan et al., 2023; Chen et al., 2023b), telesurgery (Gonzalez et al., 2023), network scheduling (Geng et al., 2020; Chen et al., 2023a), and healthcare (Yu et al., 2021).

RL applications often demand extensive training data to attain the desired accuracy level. Parallelizing training can significantly expedite this process, with one approach being Federated Reinforcement Learning (FRL) (Jin et al., 2022). In FRL, workers exchange locally trained models instead of raw data, ensuring efficient communication and data privacy. While Federated Learning (FL) is typically associated with supervised learning (Hosseinalipour et al., 2020), recent developments have extended its application to FRL, enabling multiple agents to collaboratively construct decision-making policies without sharing raw trajectories.

Distributed systems, including FRL, face vulnerabilities such as random failures or adversarial attacks. These issues may arise from agents' arbitrary behavior due to hardware glitches, inaccurate training data, or deliberate malicious attacks. In the case of attacks, the possibility exists that attackers possess extensive knowledge and can collaborate with others to maximize disruption. These scenarios fall under the Byzantine failure model, the most rigorous fault framework in distributed computing, where a minority of agents can

act arbitrarily and potentially maliciously with the aim of disrupting the system's convergence (Lamport et al., 1982). In this paper, we investigate how adversaries affect the overall convergence of federated policy gradient methods.

**Related Works:**

**1. Global Convergence of Policy Gradient Approaches:** Recently, there has been an increasing research emphasis on exploring the global convergence of PG-based methods, going beyond the well-recognized convergence to first-order stationary policies. In (Agarwal et al., 2021a), a fairly comprehensive characterization of global convergence for PG approaches is provided. Additionally, other significant studies on sample complexity for global convergence include (Wang et al., 2019; Xu et al., 2019; Liu et al., 2020; Masiha et al., 2022; Fatkhullin et al., 2023; Mondal & Aggarwal, 2024). We note that both (Fatkhullin et al., 2023) and (Mondal & Aggarwal, 2024) achieve near-optimal sample complexity of $\tilde{O}(1/\epsilon^2)$. In the absence of adversarial elements, we use the work in (Fatkhullin et al., 2023) as a foundational reference for this paper.

**2. Federated Reinforcement Learning:** Federated reinforcement learning has been explored in various setups, including tabular RL (Agarwal et al., 2021b; Jin et al., 2022; Khodadadian et al., 2022), control tasks (Wang et al., 2023b), and value-function based algorithms (Wang et al., 2023a; Xie & Song, 2023), showcasing linear speedup. In the case of policy-gradient based algorithms, it's evident that linear speedup is achievable, as each agent's collected trajectories can be parallelized (Lan et al., 2023). Nevertheless, achieving speedup becomes challenging with increasing nodes when adversaries are introduced, and this paper focuses on addressing this key issue.

3. **Byzantine Fault Tolerance in Federated Learning:** A significant body of research has focused on distributed Stochastic Gradient Descent (SGD) with adversaries, with various works such as (Chen et al., 2017; El Mhamdi et al., 2018; Yin et al., 2018; Xie et al., 2018; Alistarh et al., 2018; Diakonikolas et al., 2019; Allen-Zhu et al., 2020; Prasad et al., 2020). Most studies typically employ an aggregator to merge gradient estimates from workers while filtering out unreliable ones (Xie et al., 2018; Chen et al., 2017; El Mhamdi et al., 2018; Yin et al., 2018). However, many of these approaches make stringent assumptions regarding gradient noise, making it challenging to compare different aggregators. Notably, (Farhadkhani et al., 2022) introduces the concept of $(f, \lambda)$-resilient averaging, demonstrating that several well-known aggregators can be seen as special instances of this idea. In our work, we explore $(f, \lambda)$-averaging for combining policy gradient estimates from workers.

4. **Byzantine Fault Tolerance in Distributed Reinforcement Learning:**. We note that (Fan et al., 2021) introduces an SVRG-PG based algorithm with a local sample complexity bound of $\mathcal{O}\left(\frac{1}{N^{2/3}\varepsilon^{5/3}} + \frac{\alpha^{4/3}}{\varepsilon^{5/3}}\right)$, where $N$ represents the number of workers and $\alpha$ denotes the fraction of adversarial workers. However, they do not provide global convergence guarantees and require that the variance of importance sampling weights is upper bounded, which is unverifiable in practice. Moreover, their result is sub-optimal in $N$, which fails to provide linear speedup even when no adversaries are present. They additionally perform variance reduction with samples drawn from the central server, assuming its reliability. This is not generally allowed under federated learning since this could lead to data leakage and privacy issues (Kairouz et al., 2021). In contrast, our study centers on the global convergence of federated RL while keeping the server process minimal, aggregating different gradients using $(f, \lambda)$-resilient aggregator (Farhadkhani et al., 2022), without requiring additional samples at server.

Distributed RL algorithms with adversaries have been studied for episodic tabular MDPs in (Chen et al., 2023c). Another line of works focus on empirical evaluations of Federated RL with adversaries (Lin & Ling, 2022; Rjoub et al., 2022; Zhang et al., 2022; Xu et al., 2022) and do not provide sample complexity guarantees.

**Main Contributions:** In this paper, we address the following fundamental question:

*What is the influence of adversaries on the global convergence sample complexity of Federated Reinforcement Learning (FRL)?*

To tackle this question, we introduce Resilient Normalized Hessian Aided Recursive Policy Gradient (Res-NHARPG). Res-NHARPG integrates resilient averaging with variance-reduced policy gradient. Resilient

averaging combines gradient estimates in a manner that minimizes the impact of adversaries on the algorithm's performance, while variance reduction accelerates convergence. Our main result is that the global sample complexities of Res-NHARPG with $(f, \lambda)$ averaging is $\mathcal{O}\left(\frac{1}{\epsilon^2} \log\left(\frac{1}{\epsilon}\right)\left(\frac{1}{N} + \lambda^2 \log(N)\right)\right)$. Here, $N$ denotes the number of workers and $f < N/2$ denotes the number of faulty workers. The significance of our contributions can be summarized as follows.

1. This paper provides the first global convergence sample complexity findings for federated policy gradient-based approaches with general parametrization in the presence of adversaries.

2. We derive the sample complexity for Res-NHARPG for a broad class of aggregators, called $(f, \lambda)$-aggregators (Farhadkhani et al., 2022). This includes several popular methods such as Krum (Blanchard et al., 2017), Co-ordinate Wise Median (CWMed) and Co-ordinate Wise Trimmed Mean (CWTM) (Yin et al., 2018), Minimum Diameter Averaging (MDA) (El Mhamdi et al., 2018), Geometric Median (Chen et al., 2017) and Mean around Medians (MeaMed) (Xie et al., 2018).

3. We observe that for certain choices of aggregators (MDA, CWTM, MeaMed), our proposed approach achieves the optimal sample complexity of $\tilde{\mathcal{O}}\left(\frac{1}{N\epsilon^2}\left(1 + \frac{f^2}{N}\right)\right)$, where $\tilde{\mathcal{O}}$ ignores logarithmic factors (see Remark 4.7). In particular, this implies linear speedup when $f = \mathcal{O}(N^\delta)$, where $\delta \leq 0.5$.

We also provide experimental results showing the effectiveness of our algorithm under popular adversarial attacks (random noise, random action and sign flipping) under different environments (Cartpole-v1 from OpenAI Gym and InvertedPendulum-v2, HalfCheetah, Hopper, Inverted Double Pendulum and Walker from MuJoCo).

We obtain a significant improvement over the original bounds for $(f, \lambda)$-resilient averaging in the context of stochastic optimization (Farhadkhani et al., 2022). We achieve this by showing that our policy gradient update direction has certain desirable properties that allows us to utilize sharper concentration inequalities, instead of a simple union-bound (see Section 5).

## 2 Problem Setup

We consider a discounted Markov decision process defined by the tuple $(\mathcal{S}, \mathcal{A}, \mathcal{P}, r, \gamma)$, where $\mathcal{S}$ denotes the state space, $\mathcal{A}$ denotes the action space, $\mathcal{P}(s'|s, a)$ is the probability of transitioning from state $s$ to state $s'$ after taking action $a$, $\gamma \in (0, 1)$ is the discount factor, and $r : \mathcal{S} \times \mathcal{A} \to [-R, R]$ is the reward function of $s$ and $a$. At each time $t$, the agent is at the current state $s_t \in \mathcal{S}$ and takes action $a_t \in \mathcal{A}$ based on a possibly stochastic policy $\pi : \mathcal{S} \to \mathcal{P}(\mathcal{A})$, i.e., $a_t \sim \pi(\cdot|s_t)$. The agent then obtains a reward $r_t = r(s_t, a_t)$. The sequence of state-action pairs $\tau = \{s_0, a_0, s_1, a_1, s_2, a_2, \cdots\}$ is called a trajectory.

We consider a parameter-server architecture with a trusted central server and $N$ distributed workers or agents, labelled $1, 2, \cdots, N$. We assume each agent is given an independent and identical copy of the MDP. Among these $N$ agents, we assume that $f$ are adversarial, with these agents providing any arbitrary information (such adversaries are also called Byzantine adversaries). The aim is for the agents to collaborate using a federated policy gradient based approach to come up with a policy $\pi$ that maximizes the value function, i.e.,

$$\max_\pi \quad J(\pi) := \mathbb{E}_{s_0 \sim \rho, a_t \sim \pi(\cdot|s_t), s_{t+1} \sim \mathcal{P}(\cdot|s_t, a_t)}\left[\sum_{t=0}^\infty \gamma^t r_t\right], \tag{1}$$

where the initial state $s_0$ is drawn from some distribution $\rho$. In practice, the state and action spaces are typically very large and thus the policy is parameterized by some $\theta \in \mathbb{R}^d$. The problem in equation 1 now becomes finding $\theta$ using a collaborative approach among agents that maximizes $J(\pi_\theta)$. However since this maximization problem is usually nonconvex, it is difficult to find the globally optimal policy.

Let $J(\theta)$ denote $J(\pi_\theta)$ henceforth and let $J^* = \max_\pi J(\pi)$. The optimization problem equation 1 can be solved using the Policy Gradient (PG) approach at each agent, where the gradient $\nabla J(\theta)$ is estimated through sampling trajectories at that agent and observing rewards collected and then used in gradient ascent.

Using the current estimate of the parameter $\theta$, we aim to find $\nabla J(\theta)$ using the sampled trajectories at the agent. Let $\mathcal{G} \subset \{1, 2, \cdots, N\}$ be the set of good workers of size $N - f$. For $n \in \mathcal{G}$, let $\tau^{(n)} = \{s_0^{(n)}, a_0^{(n)}, s_1^{(n)}, \cdots, a_{H-1}^{(n)}, s_H^{(n)}\}$ be a trajectory sampled by agent $n$ under policy $\pi_\theta$ of length $H$ and $\mathcal{R}(\tau^{(n)}) = \sum_{h=0}^{H} \gamma^h r(s_h^{(n)}, a_h^{(n)})$ be the sample return. We denote the distribution of a trajectory $\tau$ of length $H$ induced by policy $\pi_\theta$ with initial state distribution $\rho$ as $p_\rho^H(\tau|\theta)$, which can be expressed as $p_\rho^H(\tau|\theta) = \rho(s_0) \prod_{h=0}^{H-1} \pi_\theta(a_h|s_h) \mathcal{P}(s_{h+1}|s_h, a_h)$. The gradient of the truncated expected return function $J_H(\theta) := \mathbb{E}_{s_0 \sim \rho, a_t \sim \pi_\theta(\cdot|s_t), s_{t+1} \sim \mathcal{P}(\cdot|s_t, a_t)}[\mathcal{R}(\tau)]$ can then be written as

$$\nabla J_H(\theta) = \int_\tau \mathcal{R}(\tau) \nabla p_\rho^H(\tau|\theta) d\tau = \int_\tau \mathcal{R}(\tau) \frac{\nabla p_\rho^H(\tau|\theta)}{p_\rho^H(\tau|\theta)} p_\rho^H(\tau|\theta) d\tau = \mathbb{E}[\mathcal{R}(\tau) \nabla \log p_\rho^H(\tau|\theta)].$$

REINFORCE (Williams, 1992) and GPOMDP (Baxter & Bartlett, 2001) are commonly used estimators for the gradient $\nabla J(\theta)$. In this work, we will be using the GPOMDP estimator given below:

$$g(\tau^{(n)}, \theta) = \sum_{h=0}^{H} \left( \sum_{t=0}^{h} \nabla_\theta \log \pi_\theta(a_t^{(n)}|s_t^{(n)}) \right) \gamma^h r(s_h^{(n)}, a_h^{(n)}). \tag{2}$$

The above expression is an unbiased stochastic estimate of $\nabla J_H(\theta)$ (Baxter & Bartlett, 2001).

## 3 Proposed Algorithm

In Res-NHARPG (given in Algorithm 1), each good agent $n \in \mathcal{G}$ computes a variance-reduced gradient estimator, $d_t^{(n)}$, in a recursive manner as follows:

$$d_t^{(n)} = (1 - \eta_t)(d_{t-1}^{(n)} + B(\hat{\tau}_t^{(n)}, \hat{\theta}_t^{(n)})(\theta_t - \theta_{t-1})) + \eta_t g(\tau_t^{(n)}, \theta_t), \tag{3}$$

where $\{\eta_t\}_t$ denotes a suitable choice of parameters, $B(\tau, \theta) := \nabla \Phi(\tau, \theta) \nabla \log p(\tau|\pi_\theta)^T + \nabla^2 \Phi(\tau, \theta)$ with $\Phi(\tau, \theta) := \sum_{t=0}^{H-1} \left( \sum_{h=t}^{H-1} \gamma^h r(s_h, a_h) \right) \log \pi_\theta(a_t, s_t)$, $\tau_t^{(n)} \sim p_\rho^H(\cdot|\pi_{\theta_t})$, $\hat{\tau}_t^{(n)} \sim p_\rho^H(\cdot|\pi_{\hat{\theta}_t^{(n)}})$, and $\hat{\theta}_t^{(n)} = q_t^{(n)} \theta_t + (1 - q_t^{(n)})\theta_{t-1}$ where $q_t^{(n)}$ is sampled from $\mathcal{U}([0, 1])$.

The update in equation 3 is inspired by the N-HARPG algorithm in (Fatkhullin et al., 2023) since its use of Hessian information was shown to help achieve sample complexity of order $\mathcal{O}\left(\frac{1}{\epsilon^2}\right)$, which is currently the state of the art. The algorithm draws inspiration from the STORM variance reduction technique (Cutkosky & Orabona, 2019), but instead of relying on the difference between successive stochastic gradients, it incorporates second-order information (Tran & Cutkosky, 2022). This approach removes the dependency on Importance Sampling (IS) and circumvents the need for unverifiable assumptions to bound the IS weights (Fatkhullin et al., 2023). In Step 10 of Algorithm 1, the update direction $d_t^{(n)}$ is computed by adding a second-order correction, $(1 - \eta_t)v_t^{(n)}$ (as defined in Step 9), to the momentum stochastic gradient $(1 - \eta_t)d_{t-1}^{(n)} + \eta_t g(\tau_t^{(n)}, \theta_t)$. The uniform sampling procedure in Steps 6-9 ensures that $v_t^{(n)}$ is an unbiased estimator of $\nabla J_H(\theta_t) - \nabla J_H(\theta_{t-1})$, closely resembling the term used in the original STORM method (Cutkosky & Orabona, 2019).

---

**Algorithm 1** Resilient Normalized Hessian-Aided Recursive Policy Gradient (Res-NHARPG)

---

1: **Input:** $\theta_0$, $\theta_1$, $d_0$, $T$, $\{\eta_t\}_{t \geq 1}$, $\{\gamma_t\}_{t \geq 1}$
2: **for** $t = 1, \ldots, T - 1$ **do**
3:    ▷ Server broadcasts $\theta_t$ to all agents
4:    ▷ Agent update
5:    **for** each agent $n \in [N]$ **do** in parallel
6:        $q_t^{(n)} \sim \mathcal{U}([0, 1])$
7:        $\hat{\theta}_t^{(n)} = q_t^{(n)} \theta_t + (1 - q_t^{(n)})\theta_{t-1}$
8:        $\tau_t^{(n)} \sim p_\rho^H(\cdot|\pi_{\theta_t}); \hat{\tau}_t^{(n)} \sim p(\cdot|\pi_{\hat{\theta}_t^{(n)}})$
9:        $v_t^{(n)} = B(\hat{\tau}_t^{(n)}, \hat{\theta}_t^{(n)})(\theta_t - \theta_{t-1})$
10:       $d_t^{(n)} = (1 - \eta_t)(d_{t-1}^{(n)} + v_t^{(n)}) + \eta_t g(\tau_t^{(n)}, \theta_t)$
11:   **end for**
12:   ▷ Server Update
13:   $d_t = F(d_t^{(1)}, \cdots, d_t^{(N)})$
14:   $\theta_{t+1} = \theta_t + \gamma_t \frac{d_t}{\|d_t\|}$
15: **end for**
16: **return** $\theta_T$

---

In Step 9, we do not need to compute and store $B(\tau, \theta)$ but only a term of form $B(\tau, \theta)u$ which can be easily computed via automatic differentiation of the scalar quantity $\langle g(\tau, \theta), u \rangle$ (Fatkhullin et al., 2023). This allows us to exploit curvature information from the policy Hessian without compromising the per-iteration computation cost.

**Aggregation at the Server:** At each iteration $t$, a good agent always sends its computed estimate back to the server, while an adversarial agent may return any arbitrary vector. At the server, we aim to mitigate the effect of adversaries is by using an aggregator to combine estimates from all agents such that bad estimates are filtered out.

The server receives $d_t^{(n)}$, $n = 1, 2, \cdots, N$ and uses the following update

$$d_t = F(d_t^{(1)}, d_t^{(2)}, \cdots, d_t^{(N)}), \text{ and} \tag{4}$$

$$\theta_{k+1} = \theta_k + \gamma_t \frac{d_t}{\|d_t\|}, \tag{5}$$

where $F$ is an aggregator and $\gamma_t > 0$ is the stepsize. For the robust aggregator $F$, we consider $(f, \lambda)$-averaging introduced in (Farhadkhani et al., 2022) as this family of aggregators encompasses several popularly used methods in Byzantine literature. The aggregator function aims to make our algorithm resilient to adversaries. The last step of the algorithm (Step 14) uses the direction $d_t$ with normalization to update the parameter $\theta_{t+1}$.

We now define $(f, \lambda)$-resilient averaging as in (Farhadkhani et al., 2022) below for ease of reference, where $f < N/2$. We note that $f < N/2$ is the optimal breakdown point as no aggregator can possibly tolerate $f \geq N/2$ adversaries (Karimireddy et al., 2020).

**Definition 3.1** $((f, \lambda)$-**Resilient averaging**)**.** For $f < N/2$ and real value $\lambda \geq 0$, an aggregation rule $F$ is called $(f, \lambda)$-*resilient averaging* if for any collection of $N$ vectors $x_1, \ldots, x_N$, and any set $\mathcal{G} \subseteq \{1, \ldots, N\}$ of size $N - f$,

$$\|F(x_1, \ldots, x_N) - \overline{x}_{\mathcal{G}}\| \leq \lambda \max_{i,j \in \mathcal{G}} \|x_i - x_j\|$$

where $\overline{x}_{\mathcal{G}} \coloneqq \frac{1}{|\mathcal{G}|} \sum_{i \in \mathcal{G}} x_i$, and $|\mathcal{G}|$ is the cardinality of $\mathcal{G}$.

Several well-known aggregators such as Krum, CWMed, CWTM, MDA, GM and MeaMed are known to be $(f, \lambda)$-resilient. We note that for these aggregators, $\lambda$ as a function of $f$ is summarized in Table 1, where these values are from (Farhadkhani et al., 2022). As a result, our unified analysis provides guarantees for all the above mentioned methods. We define and discuss these aggregators in Appendix B.

## 4 Assumptions and Main Result

In this section, we first introduce a few notations and list the assumptions used for our results. Let $d_\rho^{\pi_\theta} \in \mathcal{P}(\mathcal{S})$ denote the state visitation measure induced by policy $\pi_\theta$ and initial distribution $\rho$ defined as

$$d_\rho^{\pi_\theta}(s) \coloneqq (1 - \gamma) \mathbb{E}_{s_0 \sim \rho} \sum_{t=0}^{\infty} \gamma^t \mathbb{P}(s_t = s | s_0, \pi_\theta) \tag{6}$$

and $\nu_\rho^{\pi_\theta}(s, a) \coloneqq d_\rho^{\pi_\theta}(s) \pi(a|s)$ be the state-action visitation measure induced by $\pi_\theta$.

We assume that the policy parametrization $\pi_\theta$ is a good function approximator, which is measured by the *transferred compatible function approximation error*. This assumption is commonly used in obtaining global bounds with parametrization (Agarwal et al., 2021a; Liu et al., 2020).

**Assumption 4.1.** For any $\theta \in \mathbb{R}^d$, the *transferred compatible function approximation error*, $L_{\nu^\star}(w_\star^\theta; \theta)$, satisfies

$$L_{\nu^\star}(w_\star^\theta; \theta) \coloneqq \mathbb{E}_{(s,a) \sim \nu^\star} \left[ \left( A^{\pi_\theta}(s, a) - (1 - \gamma)(w_\star^\theta)^\top \nabla_\theta \log \pi_\theta(a|s) \right)^2 \right] \leq \varepsilon_{\text{bias}}, \tag{7}$$

| Aggregator | Computational Complexity | $\lambda$ | Sample Complexity of Res-NHARPG | Order Optimal? |
|---|---|---|---|---|
| MDA | NP-Hard | $\frac{2f}{N-f}$ | $\mathcal{O}\left(\frac{1}{\epsilon^2}\log\left(\frac{1}{\epsilon}\right)\left(\frac{1}{N}+\frac{f^2\log(N)}{N^2}\right)\right)$ | Yes |
| CWTM | $\Theta(dN)$ | $\frac{f}{N-f}\Delta$ | $\mathcal{O}\left(\frac{1}{\epsilon^2}\log\left(\frac{1}{\epsilon}\right)\left(\frac{1}{N}+\frac{f^2\Delta^2\log(N)}{N^2}\right)\right)$ | Yes |
| MeaMed | $\Theta(dN)$ | $\frac{2f}{N-f}\Delta$ | $\mathcal{O}\left(\frac{1}{\epsilon^2}\log\left(\frac{1}{\epsilon}\right)\left(\frac{1}{N}+\frac{f^2\Delta^2\log(N)}{N^2}\right)\right)$ | Yes |
| CWMed | $\Theta(dN)$ | $\frac{N}{2(N-f)}\Delta$ | $\mathcal{O}\left(\frac{1}{\epsilon^2}\log\left(\frac{1}{\epsilon}\right)\left(\frac{1}{N}+\Delta^2\log(N)\right)\right)$ | No |
| Krum | $\Theta(dN^2)$ | $1+\sqrt{\frac{N-f}{N-2f}}$ | $\mathcal{O}\left(\frac{1}{\epsilon^2}\log\left(\frac{1}{\epsilon}\right)\left(\frac{1}{N}+\frac{N-f}{N-2f}\log(N)\right)\right)$ | No |
| GM | $\mathcal{O}(dN\log^3(N/\epsilon))$ [1] | $1+\sqrt{\frac{(N-f)^2}{N(N-2f)}}$ | $\mathcal{O}\left(\frac{1}{\epsilon^2}\log\left(\frac{1}{\epsilon}\right)\left(\frac{1}{N}+\frac{N-f}{N-2f}\log(N)\right)\right)$ | No |

Table 1: In the above table, $\Delta = \min\{\sqrt{d}, 2\sqrt{N-f}\}$. The computational complexity of the aggregator, and the order optimality of sample complexity (in terms of $f$, $N$, and $\epsilon$) are also mentioned in the Table. Remark 4.8 provides a discussion on the sample complexity bounds. [1] We note that while GM is convex optimization with no closed-form solution, an $(1+\epsilon)$-approximate solution can be found in $\mathcal{O}(dN\log^3(N/\epsilon))$ time (Cohen et al., 2016).

where $A^{\pi_\theta}(s,a)$ is the advantage function of policy $\pi_\theta$ at $(s,a)$, $\nu^\star(s,a) = d_\rho^{\pi^\star}(s) \cdot \pi^\star(a|s)$ is the state-action distribution induced by an optimal policy $\pi^\star$ that maximizes $J(\pi)$, and $w_\star^\theta$ is the exact Natural Policy Gradient update direction at $\theta$.

$\varepsilon_{\text{bias}}$ captures the parametrization capacity of $\pi_\theta$. For $\pi_\theta$ using the softmax parametrization, we have $\varepsilon_{\text{bias}} = 0$ (Agarwal et al., 2021a). When $\pi_\theta$ is a restricted parametrization, which may not contain all stochastic policies, we have $\varepsilon_{\text{bias}} > 0$. It is known that $\varepsilon_{\text{bias}}$ is very small when rich neural parametrizations are used (Wang et al., 2019).

**Assumption 4.2.** For all $\theta \in \mathbb{R}^d$, the Fisher information matrix induced by policy $\pi_\theta$ and initial state distribution $\rho$ satisfies

$$F_\rho(\theta) \coloneqq \mathbb{E}_{(s,a)\sim\nu_\rho^{\pi_\theta}}\left[\nabla_\theta\log\pi_\theta(a|s)\nabla_\theta\log\pi_\theta(a|s)^\top\right] \succeq \mu_F \cdot I_d,$$

for some constant $\mu_F > 0$.

**Assumption 4.3.** There exists $\sigma > 0$ such that $g(\tau,\theta)$ defined in equation 2 satisfies $\mathbb{E}\|g(\tau,\theta)-\mathbb{E}[g(\tau,\theta)]\|^2 \le \sigma^2$, for all $\theta$ and $\tau \sim p_\rho^H(\cdot|\theta)$.

**Assumption 4.4.**     1. $\|\nabla_\theta\log\pi_\theta(a|s)\| \le G_1$ for any $\theta$ and $(s,a) \in \mathcal{S} \times \mathcal{A}$.

2. $\|\nabla_\theta\log\pi_{\theta_1}(a|s) - \nabla_\theta\log\pi_{\theta_2}(a|s)\| \le G_2\|\theta_1 - \theta_2\|$ for any $\theta_1, \theta_2$ and $(s,a) \in \mathcal{S} \times \mathcal{A}$.

**Comments on Assumptions 4.2-4.4:** We would like to highlight that all the assumptions used in this work are commonly found in PG literature. We elaborate more on these assumptions below.

Assumption 4.2 requires that the eigenvalues of the Fisher information matrix can be bounded from below. This assumption is also known as the *Fisher non-degenerate policy* assumption and is commonly used in obtaining global complexity bounds for PG based methods (Liu et al., 2020; Zhang et al., 2021; Bai et al., 2022; Fatkhullin et al., 2023).

Assumption 4.3 requires that the variance of the PG estimator must be bounded and 4.4 requires that the score function is bounded and Lipschitz continuous. Both assumptions are widely used in the analysis of PG based methods (Liu et al., 2020; Agarwal et al., 2021a; Papini et al., 2018; Xu et al., 2020; 2019; Fatkhullin et al., 2023).

Assumptions 4.2-4.4 were shown to hold for various examples recently including Gaussian policies with linearly parameterized means with clipping (Liu et al., 2020; Fatkhullin et al., 2023).

We now state the main result which gives us the last iterate global convergence rate of Algorithm 1:

**Theorem 4.5.** *Consider Algorithm 1 with $\gamma_t = \frac{6G_1}{\mu_F(t+2)}$, $\eta_t = \frac{1}{t}$ and $H = (1-\gamma)^{-1}\log(T+1)$. Let Assumptions 4.1, 4.2, 4.3 and 4.4 hold. Then for every $T \ge 1$ the output $\theta_T$ satisfies*

$$J^* - J(\theta_T) = \frac{\sqrt{\varepsilon_{\text{bias}}}}{1-\gamma} + \mathcal{O}\left(\sqrt{\frac{\log T}{NT}} + \lambda\sqrt{\frac{\log N + \log T}{T}}\right).$$

From Theorem 4.5, the number of trajectories required by Algorithm 1 to ensure $J^* - J(\theta_T) \leq \frac{\sqrt{\varepsilon_{\text{bias}}}}{1-\gamma} + \epsilon$ is

$$\mathcal{O}\left(\frac{1}{\epsilon^2} \log\left(\frac{1}{\epsilon}\right)\left(\frac{1}{N} + \lambda^2 \log(N)\right)\right).$$

For ease of exposition, we keep only dependence on $\epsilon$, $N$ and $\lambda$ while providing a detailed expression in equation 38. We also note that the expression implicitly reflects the dependence on $f$, as the value of $\lambda$ depends on both $f$ and the choice of the aggregator. The sample complexity for different aggregator functions are summarized in Table 1. The value of $\lambda$ in the table for each aggregator is from (Farhadkhani et al., 2022).

We now remark on the lower bound for the sample complexity. In order to do that, we will first provide the lower bound for Stochastic Gradient Descent with Adversaries in (Alistarh et al., 2018).

**Lemma 4.6** (Alistarh et al. (2018)). *For any $D$, $V$, and $\epsilon > 0$, let there exists a linear function $g$ : $[-D, D] \to \mathbb{R}$ (of Lipschitz continuity $G = \epsilon/D$) with a stochastic estimator $g_s$ such that $\mathbb{E}[g_s] = g$ and $\|\nabla g_s(x) - \nabla g(x)\| \leq V$ for all $x$ in the domain. Then, given $N$ machines, of which $f$ are adversaries, and $T$ samples from the stochastic estimator per machine, no algorithm can output $x$ so that $g(x) - g(x^*) < \epsilon$ with probability $\geq 2/3$ unless $T = \tilde{\Omega}\left(\frac{D^2 V^2}{\epsilon^2 N} + \frac{f^2 V^2 D^2}{\epsilon^2 N^2}\right)$, where $x^* = \arg\min_{x \in [-D, D]} g(x)$.*

*Remark* 4.7. Lemma 4.6 provides the lower bound for the sample complexity of SGD with adversaries such as: $T = \tilde{\Omega}\left(\frac{1}{\epsilon^2}\left(\frac{1}{N} + \frac{f^2}{N^2}\right)\right)$. We note that the lower bound function class in Lemma 4.6 may not satisfy the function class in this paper explicitly. However, we believe that since the lower bound holds for any class of functions that includes linear functions, the result should still hold with the assumptions in this paper. We also note that since $1/\epsilon^2$ is a lower bound for the centralized case (Mondal & Aggarwal, 2024), it follows that the lower bound for the distributed setup is $\tilde{\Omega}\left(\frac{1}{N\epsilon^2}\right)$ even in the absence of adversaries, which is the same as $\tilde{\Omega}\left(\frac{1}{\epsilon^2}\left(\frac{1}{N} + \frac{f^2}{N^2}\right)\right)$ when $f < \sqrt{N}$.

*Remark* 4.8. From Remark 4.7, it follows that MDA, CWTM and MeaMed achieve optimal sample complexity in terms of $\epsilon$, $N$ and $f$ (upto logarithmic factors). As a result, these methods exhibit linear speedup when the number of adversarial agents is on the order of $\mathcal{O}(\sqrt{N})$. When the number of adversarial agents is $\mathcal{O}(N^\delta)$, where $\delta > 0.5$, they achieve speedup of order $\mathcal{O}(N^{2(1-\delta)})$. It's worth noting that while MDA poses computational challenges, CWTM and MeaMed are computationally inexpensive, on par with simple averaging used in vanilla federated policy gradient.

*Remark* 4.9. Works in Byzantine literature generally use strong assumptions on the noise. For example, there are a line of works assuming vanishing variance (Blanchard et al., 2017; El Mhamdi et al., 2018; Xie et al., 2018), which does not hold even for the simplest policy parametrizations. Intuitively, when the assumptions on the noise is weakened, it becomes harder to distinguish adversarial and honest workers. In contrast, we provide these near-optimal bounds only using standard assumptions in Policy Gradient literature.

## 5   Proof Outline for Theorem 4.5

**Key step:** The main challenge in analyses with adversaries lies in bounding the difference of the update direction (in Algorithm 1, $d_t$) and the true gradient, $\nabla J_H(\theta_t)$.

We bound $\mathbb{E}\|d_t - \nabla J_H(\theta_t)\|$ as

$$\mathbb{E}\|d_t - \nabla J_H(\theta_t)\| \leq \mathbb{E}\|d_t - \bar{d}_t\| + \mathbb{E}\|\bar{d}_t - \nabla J_H(\theta_t)\|,$$

where $\bar{d}_t = \frac{1}{N-f}\sum_{n \in \mathcal{G}} d_t^{(n)}$. We note that $\mathbb{E}\|d_t - \bar{d}_t\|$ can be bounded using the definition of $(f, \lambda)$-aggregators as follows (details given in Detailed Outline):

$$\mathbb{E}\|d_t - \bar{d}_t\| \leq 2\lambda \mathbb{E}\left[\max_{i \in \mathcal{G}}\|d_t^{(i)} - \nabla J_H(\theta_t)\|\right]. \tag{8}$$

In (Farhadkhani et al., 2022), $\mathbb{E}[\max_{i \in \mathcal{G}} \|d_t^{(i)} - \nabla J_H(\theta_t)\|]$ is bounded by

$$\mathbb{E}\Big[\sum_{i \in \mathcal{G}} \|d_t^{(i)} - \nabla J_H(\theta_t)\|\Big] = (N - f)\mathbb{E}\|d_t^{(j)} - \nabla J_H(\theta_t)\|,$$

where $d_t^{(j)}$ is an unbiased estimate of $\nabla J_H(\theta_t)$ obtained from an honest agent $j \in \mathcal{G}$. This results in an extra factor of $(N - f)$. To illustrate the impact of this term, consider the scenario where $\lambda$ is of optimal order, specifically $\mathcal{O}\left(\frac{f}{N-f}\right)$ and $f = \mathcal{O}(N^\delta)$. Even with optimal $\lambda$ and with $0 < \delta \leq 0.5$, their sample complexity bounds do not achieve linear speedup. Furthermore, if the number of faulty workers $\delta > 0.5$, the convergence rate deteriorates with an increase in the number of workers $N$. More specifically, if $f = \Theta(N^{\delta+1/2})$, their sample complexity bound is $\mathcal{O}\left(\frac{N^{2\delta}}{\epsilon^2}\right)$, holding true for all $\delta > 0$.

In contrast, we show that for our algorithm, this term can be reduced to $\mathcal{O}(\log(N - f))$, from $(N - f)$. This results in significant improvement in the sample complexity bound, enabling us to obtain linear speedup when $\delta \leq 0.5$ and sample complexity of order $\mathcal{O}\left(\frac{N^{2(\delta-1)}}{\epsilon^2}\right)$ when $\delta > 0.5$, with certain choice of aggregators, showing speedup of order $\mathcal{O}(N^{2(1-\delta)})$.

We do this by showing that the update direction calculated by each worker $i$ in Algorithm 1, $d_t^{(i)}$, is a sum of bounded martingale differences. As a result, we can then invoke Azuma-Hoeffding inequality to obtain sharper bounds. This ensures that our algorithm guarantees speedup as long as $f = O(N^\delta)$, where $\delta < 1$.

**Detailed Outline:** We split the proof into three steps given below:

**1. Bounding $\mathbb{E}\|d_t - \bar{d}_t\|$:**

From the definition of $(f, \lambda)$-aggregator, we have $\|d_t - \bar{d}_t\| \leq \lambda \max_{i,j \in \mathcal{G}} \|d_t^{(i)} - d_t^{(j)}\|$. Thus,

$$\mathbb{E}\|d_t - \bar{d}_t\| \leq \lambda\mathbb{E}[\max_{i,j \in \mathcal{G}} \|d_t^{(i)} - d_t^{(j)}\|] \leq \lambda\mathbb{E}[\max_{i,j \in \mathcal{G}}(\|d_t^{(i)} - \nabla J_H(\theta_t)\| + \|d_t^{(j)} - \nabla J_H(\theta_t)\|)]$$

$$\leq 2\lambda\mathbb{E}[\max_{i \in \mathcal{G}} \|d_t^{(i)} - \nabla J_H(\theta_t)\|]. \tag{9}$$

Let $X_i := \|d_t^{(i)} - \nabla J_H(\theta_t)\|$ and $X := \max_{i \in \mathcal{G}} X_i$. We shall denote the indicator function of the event $A$ by $\mathbf{1}_A$. Then for all $\bar{\epsilon} > 0$

$$\mathbb{E}[X] = \mathbb{E}[X\mathbf{1}_{\{X \geq \bar{\epsilon}\}} + X\mathbf{1}_{\{X < \bar{\epsilon}\}}] \leq \mathbb{E}[X\mathbf{1}_{\{X \geq \bar{\epsilon}\}} + \bar{\epsilon}] \leq \mathbb{E}[C_1\mathbf{1}_{\{X \geq \bar{\epsilon}\}} + \bar{\epsilon}]$$

$$\leq C_1 \mathbb{P}(X \geq \bar{\epsilon}) + \bar{\epsilon} \leq C_1 \sum_{i \in \mathcal{G}} \mathbb{P}(X_i \geq \bar{\epsilon}) + \bar{\epsilon} = C_1(N - f) \mathbb{P}(\|d_t^{(i)} - \nabla J_H(\theta_t)\| \geq \bar{\epsilon}) + \bar{\epsilon},$$

where $C_1$ is an upper bound on $X$ (see Appendix C.3). Thus, for all $\bar{\epsilon} > 0$

$$\mathbb{E}\|d_t - \bar{d}_t\| \leq 2C_1\lambda(N - f) \mathbb{P}(\|d_t^{(i)} - \nabla J_H(\theta_t)\| \geq \bar{\epsilon}) + 2\lambda\bar{\epsilon}. \tag{10}$$

In order to bound $\mathbb{P}(\|d_t^{(i)} - \nabla J_H(\theta_t)\| \geq \bar{\epsilon})$, we make use of the following result.

**Lemma 5.1.** *Consider Algorithm 1. For all $i \in \mathcal{G}$ and $t \geq 1$, we have*

$$d_t^{(i)} - \nabla J_H(\theta_t) = \frac{1}{t}\sum_{j=1}^{t} M_j^{(i)},$$

*where $\|M_t^{(i)}\| \leq C_1$ and $\mathbb{E}[M_{t+1}^{(i)} \mid M_t^{(i)}] = 0$.*

The proof of the above lemma is given in Appendix C.3. Using Vector Azuma-Hoeffding inequality, we obtain the following bound (proof in Appendix C.4)

$$\mathbb{E}\|d_t - \bar{d}_t\| \leq 4e^2 C_1\lambda(N - f)e^{-(t+1)\bar{\epsilon}^2/2C_1^2} + 2\lambda\bar{\epsilon}. \tag{11}$$

Let $\bar{\epsilon} = \sqrt{\frac{2C_1^2 \log(N-f)(t+1)}{t+1}}$. Then,

$$\mathbb{E}\|d_t - \bar{d}_t\| \leq 4e^2 C_1 \lambda (N-f) e^{-(t+1)\bar{\epsilon}^2/2C_1^2} + 2\lambda\bar{\epsilon} = \frac{4e^2 C_1 \lambda}{t+1} + 2C_1 \lambda \sqrt{\frac{2\log(N-f)(t+1)}{t+1}}. \tag{12}$$

**2. Bounding $\mathbb{E}\|\bar{d}_t - \nabla J_H(\theta_t)\|$:**

We provide a bound for $\mathbb{E}\|\bar{d}_t - \nabla J_H(\theta_t)\|$ below. The proof can be found in Appendix C.2.

**Lemma 5.2.** *Consider Algorithm 1. For all $t \geq 1$, we have*

$$\mathbb{E}\|\bar{d}_t - \nabla J_H(\theta_t)\|^2 \leq \frac{C_2(1 + \log t)}{(N-f)t}, \tag{13}$$

*where $C_2$ is defined in Appendix C.2.*

From equation 12 and equation 13, we have

$$\mathbb{E}\|d_t - \nabla J_H(\theta_t)\| \leq \frac{4e^2 C_1 \lambda}{t+1} + 2C_1 \lambda \sqrt{\frac{2\log(N-f)(t+1)}{t+1}} + \sqrt{\frac{C_2(1 + \log t)}{(N-f)t}}. \tag{14}$$

**3. Obtaining the final bound:**

Using Lemma 7 of (Fatkhullin et al., 2023), we have

$$\begin{aligned} &J^* - J(\theta_{t+1}) \\ &\leq \left(1 - \frac{\sqrt{2\mu}\gamma_t}{3}\right)(J^* - J(\theta_t)) + \frac{\varepsilon'\gamma_t}{3} + \frac{8\gamma_t}{3}\mathbb{E}\|d_t - \nabla J_H(\theta_t)\| + \frac{L\gamma_t^2}{2} + \frac{4}{3}\gamma_t D_g \gamma^H, \end{aligned} \tag{15}$$

where $\mu = \frac{\mu_F^2}{2G_1^2}$, $\varepsilon' = \frac{\sqrt{\varepsilon_{\text{bias}}}}{\mu(1-\gamma)}$ and $D_g, L$ are defined in Appendix C.1.

Substituting equation 14 in equation 15 and unrolling the recursions gives us the following bound. The full proof can be found in Appendix C.5.

**Lemma 5.3.** *Consider Algorithm 1. Then for all $T \geq 1$*

$$\begin{aligned} J^* - J(\theta_T) \leq &\frac{\sqrt{\varepsilon_{\text{bias}}}}{(1-\gamma)} + \frac{J^* - J(\theta_0)}{(T+1)^2} + \frac{C_3}{T+1} \\ &+ \frac{32G_1}{\mu_F\sqrt{T+1}}\left(\sqrt{\frac{C_2(1+\log T)}{N-f}} + C_1\lambda\sqrt{2\log(N-f)(T+1)}\right), \end{aligned}$$

*where $C_3$ is defined in Appendix C.5.*

## 6   Evaluation

To show the effectiveness of our algorithm design (i.e., Res-NHARPG), we provide evaluation results on two commonly-used continuous control tasks: CartPole-v1 from OpenAI Gym (Brockman et al., 2016) and InvertedPendulum-v2 from MuJoCo (Todorov et al., 2012). Additional experiments on more demanding MuJoCo tasks, including HalfCheetah, Hopper, Inverted Double Pendulum, and Walker are provided in Appendix A. For each task on CartPole-v1 and InvertedPendulum-v2, there are ten workers to individually sample trajectories and compute gradients, and three of them are adversaries who would apply attacks to the learning process. Note that we do not know which worker is an adversary, so we cannot simply ignore certain gradient estimates to avoid the attacks. We simulate three types of attacks to the learning process: random noise, random action, and sign flipping.

In Figure 1, we present the learning process of the eight algorithms in two environments under three types of attacks. In each subfigure, the x-axis represents the number of sampled trajectories; the y-axis records the

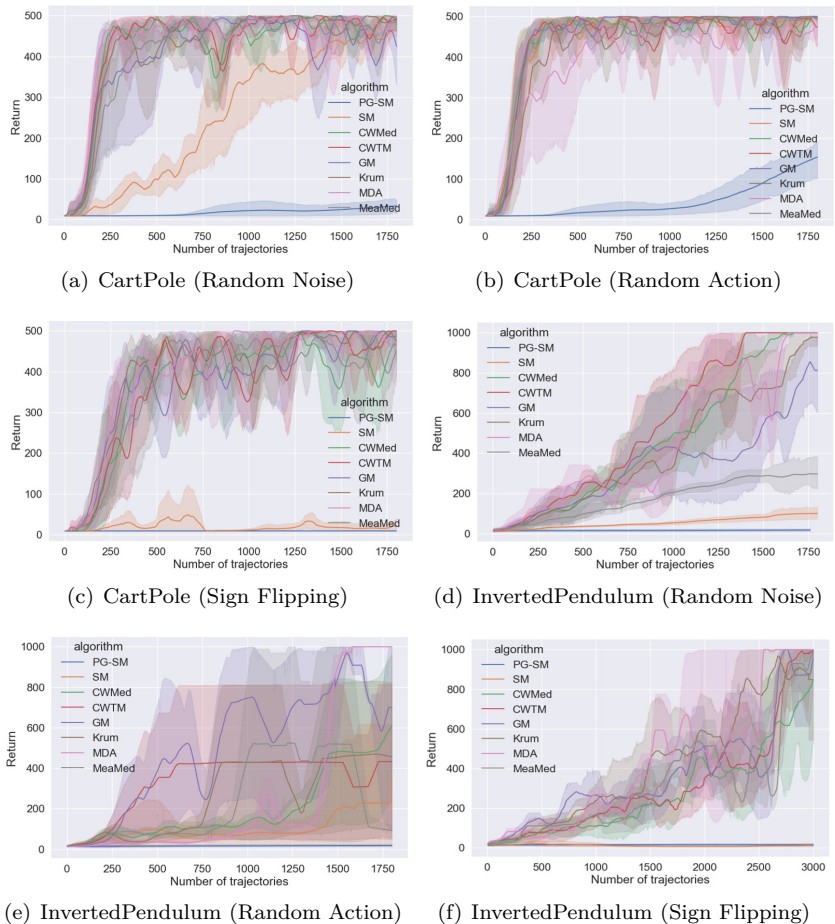

(a) CartPole (Random Noise)

(b) CartPole (Random Action)

(c) CartPole (Sign Flipping)

(d) InvertedPendulum (Random Noise)

(e) InvertedPendulum (Random Action)

(f) InvertedPendulum (Sign Flipping)

Figure 1: Evaluation results of Res-NHARPG on CartPole and InvertedPendulum. We test Res-NHARPG with the six aggregators as shown in Table 1. For baselines, we select Res-NHARPG with a simple mean (SM) function as the aggregator, which is equivalent to the original N-HARPG algorithm, and a vanilla policy gradient method with the simple mean aggregator (PG-SM). For each environment, there are ten workers, of which three are adversaries, and we simulate three types of attacks: random noise, random action, and sign flipping. It can be observed that N-HARPG outperforms PG and Res-NHARPG with those $(f, \lambda)$ aggregators can effectively handle multiple types of attacks during the learning process.

acquired trajectory return of the learned policy during evaluation. Each algorithm is repeated five times with different random seeds. The average performance and 95% confidence interval are shown as the solid line and shadow area, respectively. Codes for our experiments have been submitted as supplementary material and will be made public.

Comparing the performance of N-HARPG (i.e., SM) and Vanilla PG (i.e., PG-SM), we can see that N-HARPG consistently outperforms, especially in Figure 1(a) and 1(b) of which the task and attacks are relatively easier to deal with. For the 'random action' attack (Figure 1(b) and 1(e)), which does not directly alter the gradient estimates, N-HARPG shows better resilience. However, in more challenging tasks (e.g., InvertedPendulum) and under stronger attacks (e.g., sign flipping), both N-HARPG and Vanilla PG would likely fail, which calls for effective aggregator functions.

For CartPole, the maximum trajectory return is set as 500. Res-NHARPG, implemented with each of the six aggregators, can reach that expert level within 1000 trajectory samples, with slight difference in the convergence rate. As for InvertedPendulum, not all aggregators achieve the expert level (i.e., a trajectory return of 1000), yet they all demonstrate superior performance compared to those employing only the simple mean aggregator. It's worth noting that Res-NHARPG with the MDA aggregator consistently converges to the expert level across all test cases, showing its robustness. Moreover, the 'random action' attack brings more challenges to the aggregators, as the influence of random actions during sampling on the gradient estimates is indirect while all aggregators filter abnormal estimates based on gradient values.

## 7 Summary

In this paper, we investigate the impact of adversaries on the global convergence sample complexity of Federated Reinforcement Learning (FRL). We introduce Res-NHARPG, and show its sample complexity is of order $\tilde{\mathcal{O}}\left(\frac{1}{\epsilon^2}\left(\frac{1}{N-f}+\lambda^2\right)\right)$ using $(f,\lambda)$-aggregators, where $N$ is the total number of workers and $f$ is the number of faulty workers. Notably, when certain aggregators are used (MDA, CWTM and MeaMed), we show our approach achieves optimal sample complexity.

This work opens up multiple possible future directions in RL with adversaries, including delay in agent feedback and heterogeneous agents. We further note that a parameter free version of the proposed algorithm is also an important future direction.

## Acknowledgment

Swetha Ganesh's research is supported by the Overseas Visiting Doctoral Fellowship (OVDF) and Prime Minister's Research Fellowship (PMRF). Gugan Thoppe's research is supported by the Indo-French Centre for the Promotion of Advanced Research—CEFIPRA (7102-1), the Walmart Centre for Tech Excellence at IISc (CSR Grant WMGT-23-0001), DST-SERB's Core Research Grant (CRG/2021/008330), and the Pratiksha Trust Young Investigator Award.

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

# A    Additional experiments

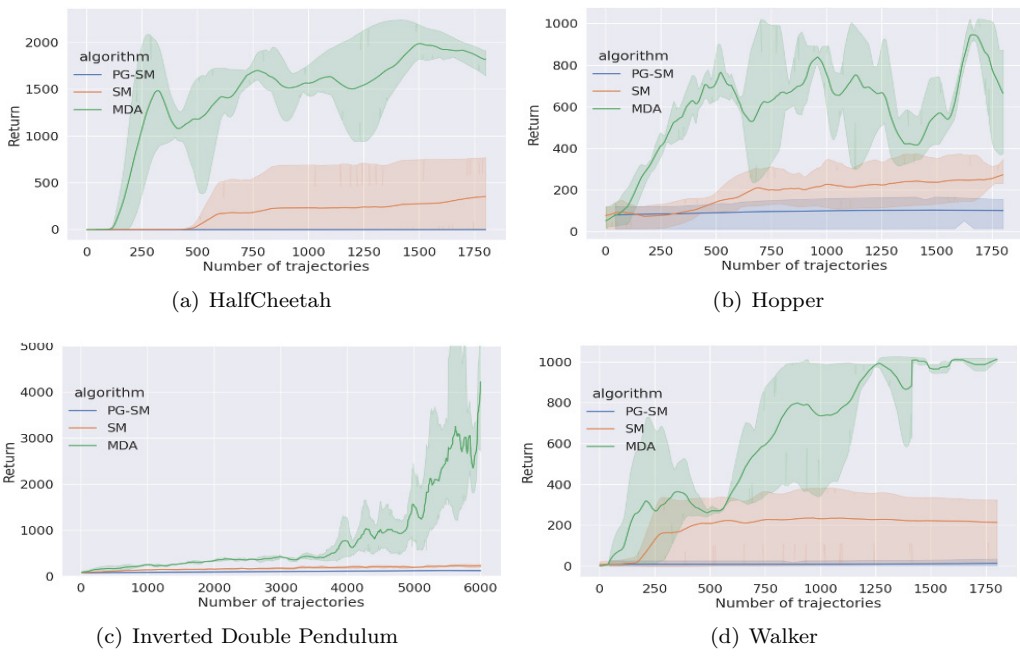

(a) HalfCheetah

(b) Hopper

(c) Inverted Double Pendulum

(d) Walker

Figure 2: Res-NHARPG with the MDA aggregator consistently outperform the baselines: N-HARPG (i.e., SM) and Vanilla PG (i.e., PG-SM), on a series of MuJoCo tasks.

Previous evaluation results have shown the superiority of Res-NHARPG with $(f, \lambda)$- aggregators. To further demonstrate its applicability, we consider Res-NHARPG with MDA and compare it with the baselines: N-HARPG (i.e., SM) and Vanilla PG (i.e., PG-SM), on a series of more challenging MuJoCo tasks: HalfCheetah, Hopper, Inverted Double Pendulum, Walker, of which the result is shown as Figure 2. Our algorithm consistently outperforms the baselines when adversaries (specifically, random noise) exist, and relatively, N-HARPG performs better than Vanilla PG. Note that our purpose is not to reach SOTA performance but to testify the effectiveness of aggregators, so the three algorithms in each subfigure share the same set of hyperparameters (without heavy fine-tuning). In Figure 2, we illustrate the training progress, up to a maximum number of sampled trajectories (6000 for the Inverted Double Pendulum and 1800 for other tasks), for each algorithm by plotting their episodic returns. The advantage of Res-NHARPG is more significant when considering the peak model performance. For instance, the highest evaluation score achieved by Res-NHARPG on Inverted Double Pendulum can exceed 9000, i.e., the SOTA performance as noted in (Weng et al., 2022), while the baselines' scores are under 500. We also evaluate the Fed-ADMM algorithm proposed in (Lan et al., 2023) in Fig. 3. As expected, even with a higher number of samples, Fed-ADMM yields significantly lower returns compared to Res-NHARPG with the MDA aggregator (see, for instance, Fed-ADMM performs poorly in HalfCheetah, achieves a return of 350 in Hopper compared to consistently above 400 for our algorithm; 90 in Inverted Double Pendulum to above 4000 for our algorithm; less than 300 in Walker to around 1000 for our algorithm).

**Details regarding attacks considered:** By 'random noise' or 'sign flipping', the real estimated policy gradients are altered by adding random noises or multiplying by a negative factor, respectively. While, for 'random action', adversarial workers would select random actions at each step, regardless of the state, when sampling trajectories for gradient estimations. Unlike the other attacks, 'random action' does not directly change the gradients, making it more challenging to detect. Also, 'random action' is different from the widely-adopted $\epsilon$-greedy exploration method, since the action choice is fully random and the randomness does not decay with the learning process.

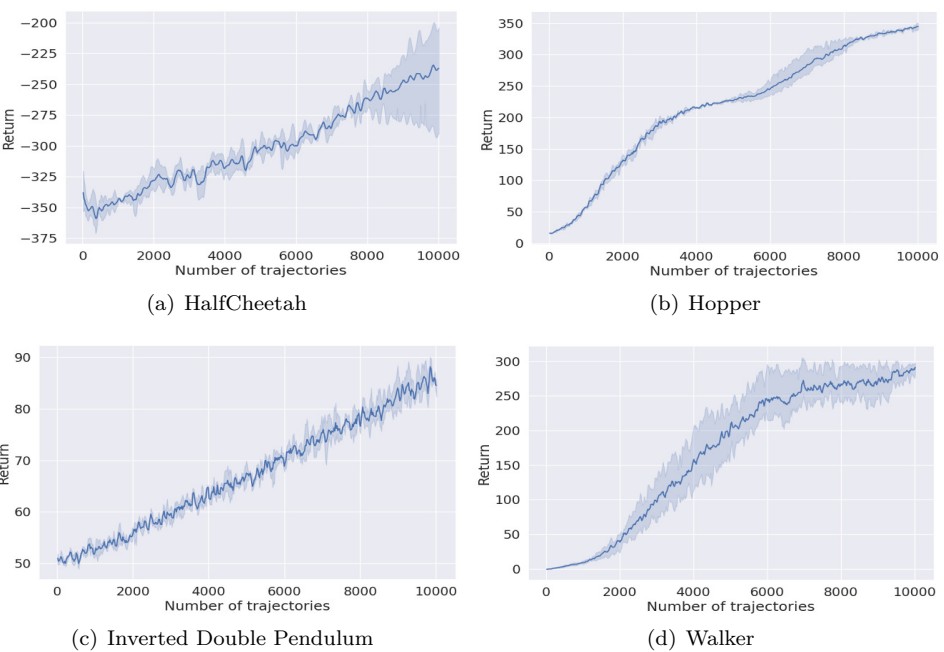

(a) HalfCheetah

(b) Hopper

(c) Inverted Double Pendulum

(d) Walker

Figure 3: Evaluation of Fed-ADMM (Lan et al., 2023) on MuJoCo tasks with random noise. The solid lines represent the mean performance, while the shaded areas indicate the 95% confidence intervals from repeated experiments. We used the official implementation from (Lan et al., 2023).

## B  Details of aggregator functions

Let $[x]_k$ denote the $k$-th coordinate of $x \in \mathbb{R}^d$. Given a set of $n$ vectors $X = \{x_1, \ldots, x_n\}$ as input, the outputs of different aggregator functions are given below.

- **Minimum Diameter Averaging (MDA)** (originally proposed in (Rousseeuw, 1985) and reused in (El Mhamdi et al., 2018)) selects a set $\mathcal{G}^*$ of cardinality $N - f$ with the smallest diameter, i.e.,

$$\mathcal{G}^* \in \underset{\substack{\mathcal{G} \subset \{1,\ldots,N\} \\ |\mathcal{G}| = N-f}}{\arg\min} \left\{ \max_{i,j \in \mathcal{G}} \|x_i - x_j\| \right\}$$

  and outputs $\frac{1}{N-f} \sum_{i \in \mathcal{G}^*} x_i$.

- **Co-ordinate wise Trimmed Mean (CWTM)** (Yin et al., 2018): Consider co-ordinate $k = 1, 2, \cdots, d$. Let $\mathcal{G}_k \subset X$ be such that $\mathcal{G}_k^c$ consists only of elements in $X$ with the $f$ largest or $f$ smallest values of $[x]_k$. Then, CWTM outputs

$$[\text{CWTM}(x_1, \ldots, x_n)]_k = \frac{1}{N - 2f} \sum_{x \in \mathcal{G}_k} [x]_k.$$

- **Co-ordinate wise Median (CWMed)** (Yin et al., 2018): The output of CWMed is given by

$$[\text{CWMed}(x_1, \ldots, x_n)]_k = \text{Median}([x_1]_k, \ldots [x_n]_k).$$

- **Mean around Median (MeaMed)** (Xie et al., 2018): MeaMed computes the average of the $N - f$ closest elements to the median in each dimension. Specifically, for each $k \in [d]$, $m \in [n]$, let $i_{m;k}$ be

the index of the input vector with $k$-th coordinate that is $m$-th closest to $\text{Median}([x_1]_k, \ldots, [x_n]_k)$. Let $C_k$ be the set of $N - f$ indices defined as

$$C_k = \{i_{1;k}, \ldots, i_{N-f;k}\}.$$

Then we have

$$[\text{MeaMed}(x_1, \ldots, x_n)]_k = \frac{1}{N-f} \sum_{i \in C_k} [x_i]_k.$$

- **Krum** (Blanchard et al., 2017): Multi-Krum* outputs an average of the vectors that are the closest to their neighbors upon discarding $f$ farthest vectors. Specifically, for each $i \in [n]$ and $k \in [n-1]$, let $i_k \in [n] \setminus \{i\}$ be the index of the $k$-th closest input vector from $x_i$, i.e., we have $\|x_i - x_{i_1}\| \leq \ldots \leq \|x_i - x_{i_{n-1}}\|$ with ties broken arbitrarily. Let $C_i$ be the set of $N - f - 1$ closest vectors to $x_i$, i.e.,

$$C_i = \{i_1, \ldots, i_{N-f-1}\}.$$

Then, for each $i \in [n]$, we define $score(i) = \sum_{j \in C_i} \|x_i - x_j\|^2$. Finally, Multi-Krum$_q^*$ outputs the average of $q$ input vectors with the smallest scores, i.e.,

$$\text{Multi-Krum}_q^*(x_1, \ldots, x_n) = \frac{1}{q} \sum_{i \in S(q)} x_i,$$

where $S(q)$ is the set of $q$ vectors with the smallest scores. We call Krum* the special case of Multi-Krum$_q^*$ for $q = 1$.

- **Geometric Median (GM)** (Chen et al., 2017): For input vectors $x_1, \ldots, x_n$, their geometric median, denoted by $\text{GM}(x_1, \ldots, x_n)$, is defined to be a vector that minimizes the sum of the distances to these vectors. Specifically, we have

$$\text{GM}(x_1, \ldots, x_n) \in \underset{z \in \mathbb{R}^d}{\arg\min} \sum_{i=1}^{n} \|z - x_i\|.$$

**Comments:**

- We observe that MDA, CWTM and MeaMed achieve optimal bounds, while CWMed, Krum and GM do not. This is explained in (Farhadkhani et al., 2022) by noting that Krum, CWMed, and GM operate solely on median-based aggregation, while MDA, CWTM, and MeaMed perform an averaging step after filtering out suspicious estimates. This averaging step results in variance reduction, similar to simple averaging in vanilla distributed SGD.

- It's worth mentioning that CWMed and GM do not require any knowledge of $f$ to be implemented. On the other hand, while MDA, CWTM, MeaMed, and Krum do rely on some knowledge of $f$, they can still be implemented by substituting an upper bound for $f$ instead. In this case, our guarantees would scale based on this upper bound, instead of $f$.

## C  Proof details

### C.1  Notations

Before proceeding further, we begin with the following lemma where we also introduce a few notations.

**Lemma C.1.** *For all $\theta \in \mathbb{R}^d$ and trajectories $\tau$ of length $H$ sampled by policy $\pi_\theta$, we have*

1. *$J$ is $L$-smooth with $L := \frac{R(G_1^2 + G_2)}{(1-\gamma)^2}$.*

2. $g(\tau, \theta)$ *is an unbiased estimate of* $\nabla J_H(\theta)$ *and*

$$\|g(\tau, \theta)\| \leq \frac{G_1 R}{(1-\gamma)^2} := G_g. \tag{16}$$

3. $B(\tau, \theta)$ *is an unbiased estimate of* $\nabla^2 J_H(\theta)$ *and*

$$\|B(\tau, \theta)\| \leq \frac{G_2 G_1^2 R + LR}{(1-\gamma)^2} := G_H. \tag{17}$$

4. *For some* $0 < \sigma_H^2 \leq G_H^2$, *we have*

$$\mathbb{E}\|B_t(\tau, \theta) - \nabla^2 J_H(\theta)\|^2 \leq \sigma_H^2. \tag{18}$$

The first two statements are given in Lemma 4.2 and 4.4, (Yuan et al., 2022) and the third and fourth statements are given in Lemma 4.1 in (Shen et al., 2019). Since the algorithm makes use of the truncated gradient and hessian estimates $\nabla J_H(\theta)$ and $\nabla^2 J_H(\theta)$ instead of $\nabla J(\theta)$ and $\nabla^2 J(\theta)$, we must bound the error terms $\nabla J(\theta) - \nabla J_H(\theta)$ and $\nabla^2 J(\theta) - \nabla^2 J_H(\theta)$. It is known that these error terms vanish geometrically fast with $H$ and is stated below:

**Lemma C.2** (Lemma 3, (Masiha et al., 2022)). *Let Assumption 4.4 be satisfied, then for all* $\theta \in \mathbb{R}^d$ *and every* $H \geq 1$, *we have*

$$\|\nabla J_H(\theta) - \nabla J(\theta)\| \leq D_g \gamma^H \ , \ \|\nabla J_H^2(\theta) - \nabla J^2(\theta)\| \leq D_h \gamma^H,$$

*where* $D_g := \frac{G_1 R}{1-\gamma}\sqrt{\frac{1}{1-\gamma} + H}$ *and* $D_h := \frac{R(G_2 + G_1^2)}{1-\gamma}\left(\frac{1}{1-\gamma} + H\right)$.

## C.2 Proof of Lemma 5.2

The proof follows similarly as in (Fatkhullin et al., 2023). Here, we take into account the affect of estimates of $(N - f)$ agents versus one agent. Since each of the $d_t^{(n)}$ are updated as

$$d_t^{(n)} = (1 - \eta_t)(d_{t-1}^{(n)} + B(\hat{\tau}_t^{(n)}, \hat{\theta}_t^{(n)})(\theta_t - \theta_{t-1})) + \eta_t g(\tau_t^{(n)}, \theta_t^{(n)}), \tag{19}$$

we notice that $\bar{d}_t = \frac{1}{N-f}\sum_{n \in \mathcal{G}} d_t^{(n)}$ can be expressed as

$$\bar{d}_t = (1 - \eta_t)(\bar{d}_{t-1} + \bar{B}(\hat{\theta}_t)(\theta_t - \theta_{t-1})) + \eta_t \bar{g}(\theta_t), \tag{20}$$

where $\bar{B}(\hat{\theta}_t) := \frac{1}{N-f}\sum_{n \in \mathcal{G}} B(\hat{\tau}_t^{(n)}, \hat{\theta}_t^{(n)})$ and $\bar{g}(\theta_t) := \frac{1}{N-f}\sum_{n \in \mathcal{G}} g(\tau_t^{(n)}, \theta_t)$.

It follows that

$$\begin{aligned}
\bar{d}_t - \nabla J_H(\theta_t) &= (1 - \eta_t)(\bar{d}_{t-1} + \bar{B}(\hat{\theta}_t)(\theta_t - \theta_{t-1})) + \eta_t \bar{g}(\theta_t) - \nabla J_H(\theta_t) \\
&= (1 - \eta_t)(\bar{d}_{t-1} - \nabla J_H(\theta_{t-1}) + \bar{B}(\hat{\theta}_t)(\theta_t - \theta_{t-1}) + \nabla J_H(\theta_{t-1}) - \nabla J_H(\theta_t))) \\
&\quad + \eta_t(\bar{g}(\theta_t) - \nabla J_H(\theta_t)) \\
&= (1 - \eta_t)(\bar{d}_{t-1} - \nabla J_H(\theta_{t-1})) + (1 - \eta_t)\bar{\mathcal{W}}_t + \eta_t \bar{\mathcal{V}}_t, \tag{21}
\end{aligned}$$

where $\bar{\mathcal{W}}_t := \bar{B}(\hat{\theta}_t)(\theta_t - \theta_{t-1}) + \nabla J_H(\theta_{t-1}) - \nabla J_H(\theta_t)$ and $\bar{\mathcal{V}}_t := \bar{g}(\theta_t) - \nabla J_H(\theta_t)$. We then have the following lemma

**Lemma C.3.** *For all* $t \geq 1$, *the following statements hold*

(i) $\mathbb{E}[\bar{\mathcal{W}}_t] = 0$

(ii) $\mathbb{E}[\bar{\mathcal{V}}_t] = 0$

*(iii)* $\mathbb{E}[\|\bar{\mathcal{V}}_t\|^2] \leq \frac{\sigma^2}{N-f}$

*(iv)* $\mathbb{E}[\|\bar{\mathcal{W}}_t\|^2] \leq \frac{12}{N-f}((2L^2 + \sigma_H^2 + D_h^2 \gamma^{2H}) \cdot \gamma_t^2 + D_g^2 \gamma^{2H}).$

*Proof of Lemma C.3.* The first two statements are easy to see since $\mathbb{E}[\bar{\mathcal{V}}_t] = \frac{1}{N-f} \sum_{n \in \mathcal{G}} \mathbb{E}[\mathcal{V}_t^{(n)}]$ and $\mathbb{E}[\mathcal{V}_t^{(n)}] = 0$ for all $n \in \mathcal{G}$ from Lemma C.1. Similar arguments show that $\mathbb{E}[\bar{\mathcal{W}}_t] = 0$.

Notice that

$$\mathbb{E}[\|\bar{\mathcal{V}}_t\|^2] = \frac{1}{(N-f)^2} \sum_{n \in \mathcal{G}} \mathbb{E}[\|\mathcal{V}_t^{(n)}\|^2] \leq \frac{\sigma^2}{N-f}, \tag{22}$$

where the last inequality follows from Assumption 4.3. Similarly,

$$\mathbb{E}[\|\bar{\mathcal{W}}_t\|^2] = \frac{1}{(N-f)^2} \sum_{n \in \mathcal{G}} \mathbb{E}[\|\mathcal{W}_t^{(n)}\|^2]. \tag{23}$$

We have for all $n \in \mathcal{G}$

$$\begin{aligned}
\mathbb{E}\|\mathcal{W}_t^{(n)}\|^2 &= \mathbb{E}\|\nabla J_H(\theta_{t-1}) - \nabla J_H(\theta_t) + B(\hat{\tau}_t^{(n)}, \hat{\theta}_t^{(n)})(\theta_t - \theta_{t-1})\| \\
&\leq 6\mathbb{E}\|\nabla J_H(\theta_{t-1}) - \nabla J_H(\theta_{t-1})\|^2 + 6\mathbb{E}\|\nabla J_H(\theta_{t-1}) - \nabla J_H(\theta_t)\|^2 \\
&\quad + 6\mathbb{E}\|B(\hat{\tau}_t^{(n)}, \hat{\theta}_t^{(n)}) - \nabla^2 J_H(\hat{\theta}_t)(\theta_t - \theta_{t-1})\|^2 + 6\mathbb{E}\|\nabla^2 J_H(\hat{\theta}_t) - \nabla^2 J(\hat{\theta}_t)(\theta_t - \theta_{t-1})\|^2 \\
&\quad + 6\mathbb{E}\|\nabla^2 J(\hat{\theta}_t)(\theta_t - \theta_{t-1})\|^2 \\
&\leq 2(6L^2 + 3\sigma_H^2)\mathbb{E}\|\theta_t - \theta_{t-1}\|^2 + 12 D_g^2 \gamma^{2H} + 6 D_h^2 \gamma^{2H} \mathbb{E}\|\theta_t - \theta_{t-1}\|^2 \\
&\leq 2(6L^2 + 3\sigma_H^2) \cdot \gamma_t^2 + 12 D_g^2 \gamma^{2H} + 6 D_h^2 \gamma^{2H} \cdot \gamma_t^2 \\
&= 12((2L^2 + \sigma_H^2 + D_h^2 \gamma^{2H}) \cdot \gamma_t^2 + D_g^2 \gamma^{2H}),
\end{aligned}$$

where the last inequality follows from the fact that $\|\theta_t - \theta_{t-1}\| = \|\gamma_t \frac{d_t}{\|d_t\|}\| = \gamma_t$. $\qquad\square$

Using Lemma C.3, we obtain

$$\begin{aligned}
\mathbb{E}\|\bar{d}_t - \nabla J_H(\theta_t)\|^2 &\leq (1 - \eta_t)\mathbb{E}\|\bar{d}_{t-1} - \nabla J_H(\theta_{t-1})\|^2 \\
&\quad + \frac{1}{N-f}(2\sigma^2 \eta_t^2 + 12((2L^2 + \sigma_H^2 + D_h^2 \gamma^{2H}) \cdot \gamma_t^2 + D_g^2 \gamma^{2H})).
\end{aligned} \tag{24}$$

Let $y_t := \frac{1}{N-f}(2\sigma^2 \eta_t^2 + 12((2L^2 + \sigma_H^2 + D_h^2 \gamma^{2H}) \cdot \gamma_t^2 + D_g^2 \gamma^{2H}))$. Unrolling the above, we obtain

$$\mathbb{E}\|\bar{d}_t - \nabla J_H(\theta_t)\|^2 \leq \sum_{i=1}^t \prod_{j=i+1}^t (1 - \eta_j) y_i.$$

For $\eta_t = \frac{1}{t}$, it follows that $\prod_{k=j_0}^j (1 - \eta_k) = \frac{j_0 - 1}{j}$ and

$$\mathbb{E}\|\bar{d}_t - \nabla J_H(\theta_t)\|^2 \leq \frac{1}{t} \sum_{i=1}^t i y_i.$$

Note that for $\eta_t = \frac{1}{t}$, $\gamma_t = \frac{6G_1}{\mu_F(t+2)}$ and $H = \frac{\log(T+1)}{(1-\gamma)}$, we have $i\eta_i^2 = \frac{1}{i}$, $i\gamma_i^2 \le \frac{6G_1}{\mu_F i}$ and $i\gamma^{2H} \le \frac{1}{i}$. Thus,

$$
\begin{aligned}
\mathbb{E}\|\bar{d}_t - \nabla J_H(\theta_t)\|^2 &\le \frac{1}{(N-f)t}\sum_{i=1}^{t}\frac{2\sigma^2}{i} + 12(2L^2 + \sigma_H^2 + D_h^2\gamma^{2H})\cdot\frac{6G_1}{\mu_F i} + \frac{24D_g^2}{i} \\
&= \left(2\sigma^2 + 12(2L^2 + \sigma_H^2 + D_h^2\gamma^{2H})\cdot\frac{6G_1}{\mu_F} + 24D_g^2\right)\cdot\frac{1}{(N-f)t}\sum_{i=1}^{t}\frac{1}{i} \\
&\le \left(2\sigma^2 + 12(2L^2 + \sigma_H^2 + D_h^2\gamma^{2H})\cdot\frac{6G_1}{\mu_F} + 24D_g^2\right)\cdot\left(\frac{1+\log t}{(N-f)t}\right) \quad (25) \\
&= \frac{C_2(1+\log t)}{(N-f)t}, \quad (26)
\end{aligned}
$$

where $C_2 := 2\sigma^2 + 12(2L^2 + \sigma_H^2 + D_h^2\gamma^{2H})\cdot\frac{6G_1}{\mu_F} + 24D_g^2$.

## C.3 Proof of Lemma 5.1

Similar to equation 21, we have

$$
d_t^{(i)} - \nabla J_H(\theta_t) = (1-\eta_t)(d_{t-1}^{(i)} - \nabla J_H(\theta_{t-1})) + \eta_t\mathcal{V}_t^{(i)} + (1-\eta_t)\mathcal{W}_t^{(i)},
$$

where $\mathcal{V}_t^{(i)} = g(\tau_t^{(i)}, \theta_t) - \nabla J_H(\theta_t)$ and $\mathcal{W}_t^{(i)} = \nabla J_H(\theta_{t-1}) - \nabla J_H(\theta_t) + B(\hat{\tau}_t^{(i)}, \hat{\theta}_t)(\theta_t - \theta_{t-1})$. Let $M_t^{(i)} := \mathcal{V}_t^{(i)} + \left(\frac{1-\eta_t}{\eta_t}\right)\mathcal{W}_t^{(i)}$. Then,

$$
d_t^{(i)} - \nabla J_H(\theta_t) = (1-\eta_t)(d_{t-1}^{(i)} - \nabla J_H(\theta_{t-1})) + \eta_t M_t^{(i)}.
$$

Unrolling the above recursion gives

$$
d_t^{(i)} - \nabla J_H(\theta_t) = \sum_{j=1}^{t}\eta_j\left(\prod_{k=j+1}^{t}(1-\eta_k)\right)M_j^{(i)},
$$

where we use the convention $\prod_{k=a}^{b}\alpha_k = 1$ if $b < a$.

Let $\eta_t = \frac{1}{t}$. With this choice of $\eta_t$, it follows that $\prod_{k=j_0}^{j_0+j}(1-\eta_k) = \frac{j_0-1}{j_0+j}$ and

$$
d_t^{(i)} - \nabla J_H(\theta_t) = \sum_{j=1}^{t}\frac{1}{j}\left(\frac{j}{t}\right)M_j^{(i)} = \frac{1}{t}\sum_{j=1}^{t}M_j^{(i)}.
$$

Now we show that $M_t^{(i)}$ is bounded and forms a martingale difference sequence. We have $\|\mathcal{V}_t^{(i)}\| \le G_g$ (from Lemma C.1) and

$$
\begin{aligned}
\|\mathcal{W}_t^{(i)}\| &\le \|\nabla J_H(\theta_{t-1}) - \nabla J_H(\theta_t) + B(\hat{\tau}_t^{(i)}, \hat{\theta}_t)(\theta_t - \theta_{t-1})\| \\
&\le \|\nabla J_H(\theta_{t-1}) - \nabla J_H(\theta_t)\| + \|B(\hat{\tau}_t^{(i)}, \hat{\theta}_t)(\theta_t - \theta_{t-1})\| \\
&\le L\|\theta_t - \theta_{t-1}\| + \|B(\hat{\tau}_t^{(i)}, \hat{\theta}_t)\|\|\theta_t - \theta_{t-1}\| \le (L+G_H)\gamma_t.
\end{aligned}
$$

Thus

$$
\|M_t^{(i)}\| = \left\|\mathcal{V}_t^{(i)} + \left(\frac{1-\eta_t}{\eta_t}\right)\mathcal{W}_t^{(i)}\right\| \le G_g + \frac{\gamma_t}{\eta_t}(L+G_H) \le G_g + \frac{6G_1}{\mu_F}(L+G_H) := C_1. \quad (27)
$$

Note that this implies that

$$
\|d_t^{(i)} - \nabla J_H(\theta_t)\| \le \frac{1}{t}\sum_{j=1}^{t}\|M_j^{(i)}\| \le C_1.
$$

Define a sequence $\{\tilde{M}_k^{(i)}\}_{k\geq 0}$ such that $\tilde{M}_0^{(i)} = 0$ and $\tilde{M}_k^{(i)} = \frac{1}{C_1}\sum_{j=0}^{k-1} M_j^{(i)}$ for $k \geq 1$. Then for all $k \geq 0$

$$\|\tilde{M}_{k+1}^{(i)} - \tilde{M}_k^{(i)}\| = \|M_k^{(i)}/C_1\| \leq 1.$$

and

$$
\begin{aligned}
\mathbb{E}[\tilde{M}_{k+1}^{(i)} \mid \tilde{M}_k^{(i)}] &= \mathbb{E}\left[\frac{1}{C_1}\sum_{j=0}^{k} M_j^{(i)} \mid \tilde{M}_k^{(i)}\right] = \mathbb{E}\left[\frac{1}{C_1}\sum_{j=0}^{k-1} M_j^{(i)} + \frac{1}{C_1}\cdot M_k^{(i)} \mid \tilde{M}_k^{(i)}\right] \\
&= \mathbb{E}\left[\tilde{M}_k^{(i)} + \frac{1}{C_1}\cdot M_k^{(i)} \mid \tilde{M}_k^{(i)}\right] = \tilde{M}_k^{(i)} + \frac{1}{C_1}\mathbb{E}\left[M_k^{(i)} \mid \tilde{M}_k^{(i)}\right].
\end{aligned}
$$

Note that

$$\sigma(\tilde{M}_k^{(i)}) \subset \sigma(\theta_0, \theta_1, q_1^{(i)}, \tau_1^{(i)}, \hat{\tau}_1^{(i)}, \theta_2, \cdots, \theta_{k-1}, q_{k-1}^{(i)}, \tau_{k-1}^{(i)}, \hat{\tau}_{k-1}^{(i)}, \theta_k) := \mathcal{F}_{k-1}^{(i)}.$$

We get

$$\mathbb{E}\left[M_k^{(i)} \mid \tilde{M}_k^{(i)}\right] = \mathbb{E}\left[\mathbb{E}[M_k^{(i)} \mid \mathcal{F}_{k-1}^{(i)}] \mid \sigma(\tilde{M}_k^{(i)})\right].$$

Observe that

$$
\begin{aligned}
\mathbb{E}[M_k^{(i)} \mid \mathcal{F}_{k-1}^{(i)}] &= \mathbb{E}\left[\mathcal{V}_k^{(i)} + \left(\frac{1-\eta_k}{\eta_k}\right)\mathcal{W}_k^{(i)} \mid \mathcal{F}_{k-1}^{(i)}\right] \\
&= \mathbb{E}\left[\mathcal{V}_k^{(i)} \mid \mathcal{F}_{k-1}^{(i)}\right] + \left(\frac{1-\eta_k}{\eta_k}\right)\mathbb{E}\left[\mathcal{W}_k^{(i)} \mid \mathcal{F}_{k-1}^{(i)}\right] \\
&= 0.
\end{aligned}
$$

Thus, for all $k \geq 0$

$$\mathbb{E}[\tilde{M}_{k+1}^{(i)} \mid \tilde{M}_k^{(i)}] = \tilde{M}_k^{(i)}. \tag{28}$$

### C.4 Proof of Equation equation 11

**Lemma C.4** (Vector Azuma-Hoeffding Inequality, (Hayes, 2005)). *Let* $\mathbf{M} = (M_0, \ldots, M_n)$ *taking values in* $\mathbb{R}^d$ *be such that*

$$M_0 = 0 \ , \ \mathbb{E}[M_n \mid M_{n-1}] = M_{n-1} \ and \ \|M_n - M_{n-1}\| \leq 1.$$

*Then, for every* $\delta > 0$,

$$\mathbb{P}(\|M_n\| \geq \delta) < 2e^2 e^{-\delta^2/2n}. \tag{29}$$

The sequence $\{\tilde{M}_k^{(i)}\}_{k\geq 0}$ satisfies all properties in Lemma C.4 which gives us

$$\mathbb{P}(\|\tilde{M}_n^{(i)}\| \geq \epsilon) \leq 2e^2 e^{-\delta^2/2n}. \tag{30}$$

It follows that

$$
\begin{aligned}
\mathbb{P}\left(\|d_t^{(i)} - \nabla J_H(\theta_t)\| \geq \epsilon\right) &= \mathbb{P}\left(\left\|\frac{1}{t+1}\sum_{j=0}^{t} M_j^{(i)}\right\| \geq \epsilon\right) = \mathbb{P}\left(\frac{C_1}{t+1}\left\|\sum_{j=0}^{t}(M_j^{(i)}/C_1)\right\| \geq \epsilon\right) \\
&= \mathbb{P}\left(\frac{C_1}{t+1}\|\tilde{M}_{t+1}^{(i)}\| \geq \epsilon\right) = \mathbb{P}\left(\|\tilde{M}_{t+1}^{(i)}\| \geq (t+1)\epsilon/C_1\right) \\
&< 2e^2 e^{-(t+1)\epsilon^2/2C_1^2}.
\end{aligned} \tag{31}
$$

Plugging the above bound into equation 10 gives

$$\mathbb{E}\|d_t - \bar{d}_t\| \leq 4e^2 C_1 \lambda (N-f)e^{-(t+1)\epsilon'^2/2C_1^2} + 2\lambda\epsilon'. \tag{32}$$

## C.5 Proof of Lemma 5.3

From equation 15, we have

$$
J^* - J(\theta_{t+1})
$$
$$
\leq \left(1 - \frac{2}{t+2}\right)(J^* - J(\theta_t)) + \frac{4}{3}\gamma_t\, D_g\, \gamma^H + \frac{8\gamma_t}{3}\mathbb{E}[\|d_t - \nabla J_H(\theta_t)\|] + \frac{L\gamma_t^2}{2} + \frac{\varepsilon'\gamma_t}{3}. \tag{33}
$$

We use the following auxillary lemma to bound the above recursion

**Lemma C.5** (Lemma 12, (Fatkhullin et al., 2023)). *Let $\tau$ be a positive integer and $\{r_t\}_{t\geq 0}$ be a sequence of non-negative numbers such that*

$$
r_{t+1} \leq (1 - \alpha_t)r_t + \beta_t,
$$

*where $\{\alpha_t\}_{t\geq 0}$ and $\{\beta_t\}_{t\geq 0}$ are non-negative sequences and $\alpha_t \leq 1$ for all $t$. Then for all $t_0, T \geq 1$,*

$$
r_T \leq \frac{(t_0 + \tau - 1)^2 r_{t_0}}{(T + \tau - 1)^2} + \frac{\sum_{t=0}^{T-1}\beta_t(t+\tau)^2}{(T+\tau-1)^2}.
$$

Using Lemma C.5 on equation 33 yields

$$
J^* - J(\theta_T) \leq \frac{J^* - J(\theta_0)}{(T+1)^2} + \frac{\sum_{t=0}^{T-1}\beta_t(t+2)^2}{(T+1)^2}, \tag{34}
$$

where

$$
\beta_t := \frac{4}{3}\gamma_t\, D_g\, \gamma^H + \frac{8\gamma_t}{3}\mathbb{E}[\|d_t - \nabla J_H(\theta_t)\|] + \frac{L\gamma_t^2}{2} + \frac{\varepsilon'\gamma_t}{3}
$$
$$
\leq \frac{4}{3}\gamma_t\, D_g\, \gamma^H + \frac{16C_1\lambda\,\gamma_t}{3}\sqrt{\frac{2\log(N-f)(t+1)}{t+1}} + \frac{64e^2C_1\lambda\gamma_t}{3(t+1)}
$$
$$
+ \frac{8\gamma_t}{3}\sqrt{\frac{C_2(1+\log t)}{(N-f)t}} + \frac{L\gamma_t^2}{2} + \frac{\varepsilon'\gamma_t}{3}
$$
$$
= \left(\frac{4}{3}\gamma_t\, D_g\, \gamma^H + \frac{L\gamma_t^2}{2} + \frac{64e^2C_1\lambda\gamma_t}{3(t+1)}\right) + \left(\frac{16C_1\lambda\,\gamma_t}{3}\sqrt{\frac{2\log(N-f)(t+1)}{t+1}}\right.
$$
$$
+ \left.\frac{8\gamma_t}{3}\sqrt{\frac{C_2(1+\log t)}{(N-f)t}}\right) + \frac{\varepsilon'\gamma_t}{3}.
$$

Note that

$$
\frac{\sum_{t=0}^{T}\varepsilon'\,\gamma_t(t+2)^2}{3(T+1)^2} = \frac{\varepsilon'\sum_{t=0}^{T}6G_1(t+2)}{3\mu_F(T+1)^2}
$$
$$
= \frac{6G_1\varepsilon'}{3\mu_F}\frac{\sum_{t=0}^{T}(t+2)}{(T+1)^2}
$$
$$
= \frac{6G_1\varepsilon'}{3\mu_F}\frac{T^2+T-1}{2(T+1)^2}
$$
$$
\leq \frac{G_1\varepsilon'}{\mu_F} = \frac{\sqrt{\varepsilon_{\text{bias}}}}{(1-\gamma)}.
$$

Also, note that since $H = \frac{\log(T+1)}{1-\gamma}$, $\gamma^H \leq \frac{1}{T+1}$ and $\gamma_t = \frac{6G_1}{\mu_F(t+2)}$

$$
\sum_{t=0}^{T}\left(\frac{4\gamma_t\, D_g\, \gamma^H}{3} + \frac{L\gamma_t^2}{2} + \frac{64e^2C_1\lambda\gamma_t}{3(t+1)}\right)(t+2)^2 \leq \sum_{t=0}^{T}\frac{16G_1D_g}{\mu_F} + \frac{18G_1^2L}{\mu_F^2} + \frac{950C_1G_1\lambda}{\mu_F}
$$
$$
= (T+1)\left(\frac{16G_1D_g}{\mu_F} + \frac{18G_1^2L}{\mu_F^2} + \frac{950C_1G_1\lambda}{\mu_F}\right).
$$

It follows that

$$\frac{1}{(T+1)^2}\sum_{t=0}^{T}\left(\frac{4\,\gamma_t\,D_g\,\gamma^H}{3}+\frac{L\,\gamma_t^2}{2}+\frac{64e^2C_1\lambda\gamma_t}{3(t+1)}\right)(t+2)^2$$

$$\leq\left(\frac{16G_1D_g}{\mu_F}+\frac{18G_1^2L}{\mu_F^2}+\frac{950C_1G_1\lambda}{\mu_F}\right)\frac{1}{T+1}.$$

Finally,

$$\frac{16C_1\lambda\gamma_t}{3}\sqrt{\frac{2\log(N-f)(t+1)}{t+1}}+\frac{8\,\gamma_t}{3}\sqrt{\frac{C_2(1+\log t)}{(N-f)t}}$$

$$=\frac{16G_1}{\mu_f(t+2)}\left(C_1\lambda\sqrt{\frac{2\log(N-f)(t+1)}{t+1}}+\sqrt{\frac{C_2(1+\log t)}{(N-f)t}}\right)$$

$$\leq\frac{16G_1}{\mu_f(t+2)\sqrt{t}}\left(C_1\lambda\sqrt{2\log(N-f)(t+1)}+\sqrt{\frac{C_2(1+\log t)}{(N-f)}}\right).$$

Note that

$$\frac{1}{(T+1)^2}\sum_{t=0}^{T}\frac{(t+2)^2}{(t+2)\sqrt{t}}=\frac{1}{(T+1)^2}\sum_{t=0}^{T}\frac{t+2}{\sqrt{t}}$$

$$\leq\frac{1}{(T+1)^2}\sum_{t=0}^{T}(\sqrt{t}+2)$$

$$\leq\frac{T^{3/2}+2T}{(T+1)^2}\leq\frac{2}{(T+1)^{1/2}}.$$

Combining all of the above, we get

$$J^*-J(\theta_T)\leq\frac{\sqrt{\varepsilon_{\text{bias}}}}{(1-\gamma)}+\frac{J^*-J(\theta_0)}{(T+1)^2}+\left(\frac{16G_1D_g}{\mu_F}+\frac{18G_1^2L}{\mu_F^2}+\frac{950C_1G_1\lambda}{\mu_F}\right)\frac{1}{T+1}$$

$$+\frac{32G_1}{\mu_F\sqrt{T+1}}\left(\sqrt{\frac{C_2(1+\log T)}{(N-f)}}+C_1\lambda\sqrt{2\log(N-f)(T+1)}\right). \tag{35}$$

From the above expression, we have

$$J^*-J(\theta_T)\leq\frac{\sqrt{\varepsilon_{\text{bias}}}}{1-\gamma}+\mathcal{O}\left(\frac{G_1}{\mu_F\sqrt{T+1}}\left(\sqrt{\frac{C_2\log T}{(N-f)}}+C_1\lambda\sqrt{\log((N-f)(T+1))}\right)\right), \tag{36}$$

where $C_1=G_g+\frac{6G_1}{\mu_F}(L+G_H)$ and $C_2=2\sigma^2+12(2L^2+\sigma_H^2+D_h^2\gamma^{2H})\cdot\frac{6G_1}{\mu_F}+24D_g^2$. The Lipschitz constant $L$, variance bounds $\sigma^2$ and $\sigma_H^2$ and the remaining terms $G_g$, $G_H$, $D_h$ and $D_g$ in turn, can be bounded in terms of $\gamma$, $\mu_F$, the bound on the reward function $R$ and the Lipschitz and smoothness constants of the score function $G_1$, $G_2$ (see Lemma C.1 and C.2). Substituting these bounds, we obtain

$$J^*-J(\theta_T)\leq\frac{\sqrt{\varepsilon_{\text{bias}}}}{1-\gamma}+\mathcal{O}\left(\frac{G_1}{\mu_F\sqrt{T+1}}\left(\sqrt{\frac{\frac{G_1}{\mu_F}\cdot\left(\frac{R^2G_2^2G_1^4}{(1-\gamma)^4}+\frac{R^4(G_1^4+G_2^2)}{(1-\gamma)^8}\right)\log T}{(N-f)}}\right.\right.$$

$$\left.\left.+\left(\frac{RG_2G_1^2}{(1-\gamma)^2}+\frac{R^2(G_1^2+G_2)}{(1-\gamma)^4}\right)\lambda\sqrt{\log((N-f)(T+1))}\right)\right). \tag{37}$$

With the above bound, we obtain $J^* - J(\theta_T) \leq \frac{\sqrt{\varepsilon_{\mathrm{bias}}}}{1-\gamma} + \epsilon$ for

$$
\begin{aligned}
T = \mathcal{O}\Bigg( \frac{1}{\epsilon^2} \log\left(\frac{1}{\epsilon}\right) \Bigg( & \frac{1}{(N-f)} \cdot \left( \frac{R^2 G_2^2 G_1^7}{\mu_F^3 (1-\gamma)^4} + \frac{R^4 (G_1^7 + G_1^3 G_2^2)}{\mu_F^3 (1-\gamma)^8} \right) \\
& + \lambda^2 \log(N-f) \cdot \left( \frac{R^2 G_2^2 G_1^6}{\mu_F^2 (1-\gamma)^4} + \frac{R^4 (G_1^6 + G_1^2 G_2^2)}{\mu_F^2 (1-\gamma)^8} \right) \Bigg) \Bigg).
\end{aligned}
\tag{38}
$$

## D  Compute Resources

Experiments were conducted using the Oracle Cloud infrastructure, where each computation instance was equipped with 8 Intel Xeon Platinum CPU cores and 128 GB of memory. In Figure 1, each subfigure contains comparisons among eight algorithms each of which is repeated five times with different random seeds. The single running time for an algorithm is approximately two hours. Thus, to reproduce Figure 1, it requires approximately 480 CPU hours. Similarly, for Figure 2, the time would be about 120 CPU hours.

