# OpenReview forum: "Global Convergence Guarantees for Federated Policy Gradient Methods with Adversaries"
_TMLR — Accepted by TMLR_

### Review · Reviewer_U4B5 · 2024-08-28

**Summary Of Contributions:**

This paper designs an algorithm for FRL with adversarial agents and general policy parameterization by combining the $(f,\lambda)$-resilient averaging and the N-HARPG from previous literature. Global convergence guarantee is provided, showing resilience of the algorithm to adversaries. Empirical evaluations are also provided.

**Audience:**

Yes

**Broader Impact Concerns:**

Nnone of those are considered as necessary to be specifically highlighted here.

**Claims And Evidence:**

Yes

**Requested Changes:**

According to the weaknesses, the following points, which are critical for acceptance, should be addressed.
- Make the main message consistent. The authors may talk more about "resilience to adversaries" in the "main contribution" paragraph, in section 3 (on which parts of the algorithm design enable this resilience), and in section 4 (on how the theorem implies resilience and how it compares to previous literature).
- The authors may modify the sentences and equations that are inaccurate.
- The authors may talk more about the missing background definitions and intuitions.

The following concerns should also be address:
- The authors should discuss how to set $\gamma_t$ in practice if we don't know the required parameters, or how to estimate the required parameters.
- Correct the mistakes.

**Strengths And Weaknesses:**

**Strengths:**
- The paper studies an interesting and important problem.
- The assumptions are clearly listed and the theoretical results are clean.
- Empirical studies are used to demonstrate the effectiveness of the algorithm.

**Weaknesses:**
- The main message isn't consistent throughout the paper. From the title, abstract and conclusion, the most important contribution seems to be the algorithm's resilience to adversaries. However, this is not discussed in the "main contributions" paragraph of introduction, in section 3 (when discussing the algorithm design) and section 4 (when explaining the theoretical result).
I believe the algorithm is resilient to adversaries because the information is aggregated by $(f,\lambda)$-resilient averaging. Currently the readers can only confirm this when reading the proof sketch.
- Some messages are not accurate. For example, in the abstract, the authors claim achieving optimal sample complexity. However, from Table 1, the optimal sample complexity is achieved only when the MDA aggregator is used. The optimality is not universal across all aggregators, and the only optimal aggregator is NP-hard to compute. Another example is the sample complexity $\tilde{\mathcal{O}}(\frac{1}{\epsilon^2}\log(\frac{1}{\epsilon}(\frac{1}{N}+\lambda^2\log(N))))$ (in introduction and after theorem 4.5). The dependence of $f$ (or $N-f$) is omitted. I understand that this is neglectable when $f<N/2$. But it's very confusing if parameter $f$ is not in the result, since the aggregator is $(f,\lambda)$-resilient.
- Some background information is missing. I understand that many components of the algorithm are from existing result. But it would greatly help if the authors could talk more about the intuitions, or definitions, behind them. Examples include equation (3), line 6-10 of algorithm 1 (i.e. approximation of equation (3)), the word "Byzantine adversaries" at the beginning of section 5.
- To achieve the theoretical result, we need to set $\gamma_t$ using parameter $\mu_F$ and $G_1$. However, they may not be accessible in practice. The authors should discuss how to tackle this challenge.
- Some small mistakes. (i). "R->r" in the five-tuple of the first line at the beginning of section2. (ii). A in equation 7 is not defined. (iii). i is not defined on the RHS of the equation right below equation (8). (iv) "$J_H\theta_t)$->$J_H(\theta_t)$" in equation (9).

---

> ### Author Response · Authors · 2024-09-05
> **Response Part 1**
>
> >  The main message isn't consistent throughout the paper. From the title, abstract and conclusion, the most important contribution seems to be the algorithm's resilience to adversaries. However, this is not discussed in the "main contributions" paragraph of introduction, in section 3 (when discussing the algorithm design) and section 4 (when explaining the theoretical result). I believe the algorithm is resilient to adversaries because the information is aggregated by  (f,λ)-resilient averaging. Currently the readers can only confirm this when reading the proof sketch.
>
> Thanks a lot for your valuable comments. We agree that the primary contribution is the development of a federated reinforcement learning algorithm that functions effectively in the presence of adversaries. Below, we outline the changes or comments in each section to emphasize this message:
>
> Introduction: The contributions section starts with the question "*What is the influence of adversaries on the global convergence sample complexity of Federated Reinforcement Learning (FRL)?*" This puts the focus that the question is the influence of adversaries. Further, we added in the first paragraph of the contributions that "Res-NHARPG integrates resilient averaging with variance-reduced policy gradient. Resilient averaging combines gradient estimates in a manner that minimizes the impact of adversaries on the algorithm's performance, while variance reduction accelerates convergence." We further modified the first contribution to make it more precise - "This paper provides the first global convergence sample complexity findings for federated policy gradient-based approaches with general parametrization *in the presence of adversaries*."
>
> Section 3: We described the robust aggregator better for their role in adversaries - "For the robust aggregator $F$, we consider $(f,\lambda)$-averaging introduced in (Farhadkhani et al., 2022) as this family of aggregators encompasses several popularly used methods in Byzantine literature. The aggregator function aims to make our algorithm resilient to adversaries."
>
> Section 4: We have increased the discussion of adversaries in our work. Specifically, we clarified that the parameter $\lambda$ depends on adversaries and expanded the explanation of the lower bound to explicitly highlight their impact. Additionally, the Remarks following the Theorem offer comparisons and interpretations of the results in the context of adversaries. These revisions emphasize the resilience aspect of the lower bound in comparison to other studies. However, since no existing work addresses global optimality in federated reinforcement learning with adversaries, our comparison is limited to the lower bound.
>
>
>
> > Some messages are not accurate. For example, in the abstract, the authors claim achieving optimal sample complexity. However, from Table 1, the optimal sample complexity is achieved only when the MDA aggregator is used. The optimality is not universal across all aggregators, and the only optimal aggregator is NP-hard to compute. Another example is the sample complexity  O~(1ϵ2log⁡(1ϵ(1N+λ2log⁡(N))))  (in introduction and after theorem 4.5). The dependence of  f  (or  N−f) is omitted. I understand that this is neglectable when  f<N/2. But it's very confusing if parameter  f  is not in the result, since the aggregator is  (f,λ)-resilient.
>
> It is important to clarify that MDA is not the only method achieving optimal sample complexity. As noted, "For certain choices of aggregators (MDA, CWTM, MeaMed), our proposed approach attains the optimal sample complexity of $\tilde{O} \left( \frac{1}{N\epsilon^2} \left(1 + \frac{f^2}{N}\right)\right)$, where $\tilde{O}$ disregards logarithmic factors." We also highlight that $\Delta \le \sqrt{d}$, meaning it is a constant that does not affect the order dependence. Therefore, both CWTM and MeaMed can achieve order-optimal sample complexity without being NP-hard.
>
> Regarding the dependence omission, we note that $\lambda$ as a function of $f$ is dependent on the  choice of the aggregator. We see that in Table 1, where $\lambda$ as a function of $f$ is given for the different aggregators. In order to avoid the confusion, we added explicitly in Section 3 that "We note that for these aggregators, $\lambda$ as a function of $f$ is summarized in Table 1, where these values are from (Farhadkhani et al., 2022)." Further, we added after the statement of Theorem 4.5 that "We also note that the expression implicitly reflects the dependence on $f$, as the value of $\lambda$ depends on both $f$ and the choice of the aggregator."

---

> ### Author Response · Authors · 2024-09-05
> **Response Part 2**
>
> >Some background information is missing. I understand that many components of the algorithm are from existing result. But it would greatly help if the authors could talk more about the intuitions, or definitions, behind them. Examples include equation (3), line 6-10 of algorithm 1 (i.e. approximation of equation (3)), the word "Byzantine adversaries" at the beginning of section 5.
>
>
> We have added an explanation for the variance-reduction method following equation (3). This explanation also includes the explanation of lines 6-10 of algorithm. In particular, we added: "The update in equation 3 is inspired by the N-HARPG algorithm in (Fatkhullin et al., 2023)  since its use of Hessian information was shown to help achieve sample complexity of order $O\left(\frac{1}{\epsilon^2}\right)$, which is currently the state of the art. {The algorithm draws inspiration from the STORM variance reduction technique (Cutkosky & Orabona, 2019), but instead of relying on the difference between successive stochastic gradients, it incorporates second-order information (Tran & Cutkosky, 2022). This approach removes the dependency on Importance Sampling (IS) and circumvents the need for unverifiable assumptions to bound the IS weights (Fatkhullin et al., 2023).  In Step 10 of Algorithm 1, the update direction $d_t^{(n)}$ is computed by adding a second-order correction, $(1-\eta_t)v_t^{(n)}$ (as defined in Step 9), to the momentum stochastic gradient $(1-\eta_t)d^{(n)}_{t-1} + \eta_t g(\tau_t^{(n)}, \theta_t)$.
>
> The uniform sampling procedure in Steps 6-9 ensures that $v_t^{(n)}$ is an unbiased estimator of $\nabla J_H(\theta_t) - \nabla J_H(\theta_{t-1})$, closely resembling the term used in the original STORM method (Cutkosky & Orabona, 2019)."
>
> For the "Byzantine adversaries", we changed it to "adversaries". Also in the problem setup - we added "(such adversaries are also called Byzantine adversaries)".
>
>
> >To achieve the theoretical result, we need to set  γt  using parameter  μF  and  G1. However, they may not be accessible in practice. The authors should discuss how to tackle this challenge.
>
> Thanks for this great question. While parameter free optimization as mentioned will be an important future result, most of the results in the literature on reinforcement learning depends on such parameters and such choices are also there in the related works on reinforcement learning. We have mentioned this in the future work direction. *We further note that a parameter free version of the proposed algorithm is also an important future direction.*
>
> >Some small mistakes. (i). "R->r" in the five-tuple of the first line at the beginning of section2. (ii). A in equation 7 is not defined. (iii). i is not defined on the RHS of the equation right below equation (8). (iv) "JHθt)->JH(θt)" in equation (9).
>
> (i) We have changed $R$ to $r$ (ii) We have added the definition of $A$ below equation 7 (iii) We have updated accordingly (iv) The missing bracket is added in equation 9.
>
>
> >Make the main message consistent. The authors may talk more about "resilience to adversaries" in the "main contribution" paragraph, in section 3 (on which parts of the algorithm design enable this resilience), and in section 4 (on how the theorem implies resilience and how it compares to previous literature).  The authors may modify the sentences and equations that are inaccurate. The authors may talk more about the missing background definitions and intuitions.
>
> Thanks a lot for pointing this out. We have addressed these in the response to the pointed Weaknesses above.
>
> >The following concerns should also be address: The authors should discuss how to set  γt  in practice if we don't know the required parameters, or how to estimate the required parameters.  Correct the mistakes.
>
> We have addressed these in the response to the pointed Weaknesses above.

---

### Review · Reviewer_BVJf · 2024-08-29

**Summary Of Contributions:**

This paper studies federated reinforcement learning under Byzantine attacks. They propose an algorithm for this problem along with a global convergence sample complexity. The algorithm is based on a variance-reduced gradient estimator utilizing Hessian information and is compatible with any $(f,\lambda)-$Resilient aggregator. They achieve a sample complexity $O\left(\frac{1}{\epsilon^2}\log(\frac{1}{\epsilon})(\frac{1}{N}+\lambda^2\log(N))\right)$. And they show for some specific aggregator (i.e., some specific $\lambda$), the sample complexity is of optimal order. They also conduct experiments on continuous tasks to show the advantage of the proposed algorithm.

**Audience:**

Yes

**Broader Impact Concerns:**

There is no concerns on the ethical implications of this work.

**Claims And Evidence:**

Yes

**Requested Changes:**

Refer to the weaknesses I listed. Besides, there are some minor problems. For example, a quantify should be explained if it first appears in the manuscript ($A^{\pi_\theta}(s,a)$ in (7)). Some missing brackets ($\nabla J_H\theta_t$ in (9)).

**Strengths And Weaknesses:**

### Strengths
1. This work provides the first global convergence sample complexity for federated reinforcement learning under adversarial attacks. Their algorithm and its analysis is compatible with any $(f,\lambda)-$resilient aggregator, hence they can provide sample complexity results for a bunch of popular algorithms.
2. This work considers parameterized policy and the result includes a parametrization error characterizes the representational capacity of the policy class.
3. The assumptions are well listed and discussed.
4. Besides the theoretical guarantees, they conducted experiments which showed the advantage of the proposed algorithm.

### Weaknesses
1. In Theorem 4.5, to simplify the expression, they keep only dependence on $\epsilon, N, \lambda$. Why is $f$ omitted which I think is an important factor for this problem?
2. The discussion on the lower bound (i.e., Remark 4.6) is not clear. As one of the contributions listed is to claim the optimal sample complexity under some aggregators, I think the remark on the lower bound should be more comprehensive.
3. The theory shows for some aggregators, an optimal sample complexity is achieved while for other aggregators, it is not. This finding is not reflected obviously in the experiments. For example, in (a), CWMed (not optimal from theory) has faster convergence rate than MeaMed (optimal from theory).

---

> ### Author Response · Authors · 2024-09-05
> **Response to the comments**
>
> >In Theorem 4.5, to simplify the expression, they keep only dependence on  ϵ,N,λ. Why is  f  omitted which I think is an important factor for this problem?
>
> Thanks a lot for pointing this out. We note that $\lambda$ as a function of $f$ is dependent on the  choice of the aggregator. We see that in Table 1, where $\lambda$ as a function of $f$ is given for the different aggregators. In order to avoid the confusion, we added explicitly in Section 3 that "We note that for these aggregators, $\lambda$ as a function of $f$ is summarized in Table 1, where these values are from (Farhadkhani et al., 2022)." Further, we added after the statement of Theorem 4.5 that "We also note that the expression implicitly reflects the dependence on $f$, as the value of $\lambda$ depends on both $f$ and the choice of the aggregator."
>
> >The discussion on the lower bound (i.e., Remark 4.6) is not clear. As one of the contributions listed is to claim the optimal sample complexity under some aggregators, I think the remark on the lower bound should be more comprehensive.
>
> Thanks a lot for pointing this. We gave revised the entire discussion on the lower bound, making it formal. Please see below for the modified part.
>
> " We now remark on the lower bound  for the sample complexity. In order to do that, we will first provide the lower bound for  Stochastic Gradient Descent with Adversaries in (Alistarh et al., 2018).
> Lemma 4.6 ((Alistarh et al., 2018)).    For any $D$, $V$,  and $\epsilon>0$, let there exists a linear function $g : [-D, D] \to{\mathbb{R}}$ (of Lipschitz continuity $G = \epsilon/D$) with a stochastic estimator $g_s$ such that ${\mathbb E}[g_s]=g$ and $||\nabla g_s(x) - \nabla g(x)||\le V$ for all $x$ in the domain.  Then, given $N$ machines, of which $f$ are adversaries, and $T$ samples from the stochastic estimator per machine, no algorithm can output $x$ so that $g(x) - g(x^*) < \epsilon$ with probability  $\geq 2/3$
> unless $T = {\tilde \Omega}\left(\frac{D^2V^2}{\epsilon^2 N} + \frac{f^2V^2D^2}{\epsilon^2N^2 }\right)$, where $x^* = \arg\min_{x\in [-D,D]} g(x)$.
>
> Remark 4.7: Lemma 4.6  provides the lower bound for the sample complexity of  SGD with adversaries as:    $T = {\tilde \Omega} \left(\frac{1}{\epsilon^2} \left(\frac{1}{N} + \frac{f^2}{N^2}\right)\right).$ "
>
> >The theory shows for some aggregators, an optimal sample complexity is achieved while for other aggregators, it is not. This finding is not reflected obviously in the experiments. For example, in (a), CWMed (not optimal from theory) has faster convergence rate than MeaMed (optimal from theory).
>
> We would like to clarify the following points:
>
> 1. The results presented are in terms of order, although constants do influence the simulations. While the simulations demonstrate the desired convergence properties in the presence of adversaries, comparing the performance of different aggregators based on these results may not be ideal due to the impact of constants in the theoretical analysis.
>
> 2. The results represent achievable bounds, but they do not imply that CWMed, for example, cannot perform better. It is possible that a different analytical approach for other aggregators could yield improved results, a possibility not ruled out by this paper.
>
> Therefore, we believe it is not ideal to compare the aggregators based solely on the experiments. Instead, the experiments are intended to highlight the convergence properties of the aggregators in adversarial settings, showing that the proposed Res-NHARPG algorithm is effective in such scenarios.
>
>
> >Refer to the weaknesses I listed. Besides, there are some minor problems. For example, a quantify should be explained if it first appears in the manuscript (Aπθ(s,a)  in (7)). Some missing brackets (∇JHθt  in (9)).
>
> Thanks for pointing these issues. We have included the definition of $A^{\pi}(s,a)$ after (7) and added the missing bracket in (9).

---

### Review · Reviewer_HHRA · 2024-09-16

**Summary Of Contributions:**

The authors study policy-based federated RL with adversaries: N agents or nodes contribute to the optimization of a parametric policy for the same MDP. They do so by collecting trajectories with the same policy in independent copies of the MDP and sending policy gradient estimates to a trusted central server that updates the policy parameters by (normalized) gradient ascent and broadcasts the updated parameters to all agents (no trajectory data are shared). Among the N agents hide f adversaries that can send arbitrary values to the server.

An algorithm with optimal sample complexity (to find a globally optimal policy) is provided under standard assumptions. This is the first such result for this specific setting. Its optimality is guaranteed by a lower bound for stochastic gradient descent with adversaries.

Experiments on Mujoco tasks with 3 adversaries out of 10 nodes show the advantages compared to policy gradient algorithms that do not take into account the possible presence of adversarial agents.

**Audience:**

Yes

**Claims And Evidence:**

Yes

**Requested Changes:**

The only adjustment that I deem critical has to do with the lower bound (Lemma 4.6 and Remark 4.8). Intuitively, the lower bound does hold for your setting. However, I think that a more formal reduction argument is needed. Is it possible for the "hard" linear function by (Alistarh et al., 2018) to appear as the performance function (expected return) of a reasonable policy class in some MDP? By reasonable policy class, ideally, I mean one that satisfies your assumptions 4.1-4.4. If this is not possible, at least a discussion about this and possible mismatches between the upper and the lower bound is required. Another aspect that should be clarified is the dependence on the task horizon, that can only result from an ad-hoc lower bound for MDPs.

Regarding the experiments, the approach is not compared with federated RL algorithms from the literature. It would make sense to compare at least with (Lan et al. 2023) that is policy-based.

Minor issues:
- Assumption 4.4 does *not* hold for Gaussian policies, unless some sort of clipping is used. cf Assumption 4.1 by Yuan et al. 2022 and the related discussion
- "late-iterate" -> last iterate

**Strengths And Weaknesses:**

The paper studies an interesting (although very specific) setting and provides the first sample complexity result for it.
To ensure optimality, some care in the technical analysis was needed. The proof technique is well explained in the paper.

The contribution relies heavily on previous work from two well-established lines:
- Policy gradient with global convergence, in particular (Fatkhullin et al., 2023), which provides the standard assumptions on the policy class, the policy gradient estimates (for the agents) based on second order information and the analysis that leads to the optimal $\epsilon^{-2}$ rate
- Federated learning, providing the robust aggregation techniques that allow a linear speedup w.r.t. the number of agents even when adversaries are present

The main effort of this paper is thus combining these results into a single federated-PG algorithm for the new setting. On one hand, this means that no truly original ideas are introduced. On the other hand, as mentioned above, this combination does require some care from a purely technical standpoint. The relationship with previous works and the key technical challenges are well documented in the paper.

---

> ### Author Response · Authors · 2024-09-23
> **Response to the comments**
>
> >The only adjustment that I deem critical has to do with the lower bound (Lemma 4.6 and Remark 4.8). Intuitively, the lower bound does hold for your setting. However, I think that a more formal reduction argument is needed. Is it possible for the "hard" linear function by (Alistarh et al., 2018) to appear as the performance function (expected return) of a reasonable policy class in some MDP? By reasonable policy class, ideally, I mean one that satisfies your assumptions 4.1-4.4. If this is not possible, at least a discussion about this and possible mismatches between the upper and the lower bound is required. Another aspect that should be clarified is the dependence on the task horizon, that can only result from an ad-hoc lower bound for MDPs.
>
> We agree with the reviewer that the "hard" linear function by (Alistarh et al., 2018) may not satisfy Assumptions 4.1-4.4 in this work. We have added that in Remark 4.8, where we added "We note that the lower bound function class in Lemma 4.6 may not satisfy the function class in this paper explicitly. However, we believe that since the lower bound holds for any class of functions that includes linear functions, the result should still hold with the assumptions in this paper. We also note that since $1/\epsilon^2$ is a lower bound for the centralized case (Mondal and Aggarwal, 2024), it follows that the lower bound for the distributed setup is ${\tilde \Omega}\left(\frac{1}{N\epsilon^2}\right)$ even in the absence of adversaries, which is the same as ${\tilde \Omega} \left(\frac{1}{\epsilon^2} \left(\frac{1}{N} + \frac{f^2}{N^2}\right)\right)$ when $f<\sqrt{N}$. "
>
> Regarding the dependence on $1-\gamma$, we note that in equation (38) this dependence appears as $1/(1-\gamma)^8$ for our result. We agree that this is not explicitly included in the bounds since $\gamma$ is assumed to be constant, while it is important and thus included in (38) to ensure completeness of our result.
>
>
> >Regarding the experiments, the approach is not compared with federated RL algorithms from the literature. It would make sense to compare at least with (Lan et al. 2023) that is policy-based.
>
> Thank you for bringing this up. We note that the NPG approach in (Lan et al. 2023) is not known to achieve better sample complexity than $1/\epsilon^3$, even in a centralized setting. For this reason, we initially did not include it in our comparison with policy gradient methods. However, in the revision, we have included a comparison with the approach in (Lan et al. 2023). It is important to highlight that (Lan et al. 2023) does not account for adversaries, and, as with the policy gradient approach in our current manuscript, its performance is not good in the presence of adversaries. The new Fig. 3 in the manuscript shows the performance of the approach in (Lan et al. 2023), and corresponding writeup is added to show that our approach which is resilient to adversaries perform much better.
>
> >Minor issues:
> >-   Assumption 4.4 does  _not_  hold for Gaussian policies, unless some sort of clipping is used. cf Assumption 4.1 by Yuan et al. 2022 and the related discussion
> >-  "late-iterate" -> last iterate
>
> Thanks for pointing this out. We have explicitly added the clipping in explanation of the Assumption. Also, we fixed the typo of "late" to "last".

---

> > ### Comment · Reviewer_HHRA · 2024-10-15
> > **Thanks**
> >
> > Thank for your answers. I am satisfied with the additions.

---

### Decision · Action_Editor_CP2W · 2024-10-22

**Recommendation:** Accept as is

**Comment:**

This paper studies a special but practically relevant setting within reinforcement learning. Multiple decentralized agents are given independent copies of the same MDP. Each of them computes a policy gradient estimate, sends it to a central server that runs a policy gradient algorithm, and then receives the updated parameter from the central server. It is further assumed that among the agents some of them are adversarial, and the goal is to ensure global convergence while being resilient to the adversaries.

All reviewers found the setting interesting and praised the technical quality of the paper. The proposed algorithm and its analysis combine a few known components from the literature (policy gradient methods and federated learning), but materializing it requires special technical care, which the paper does very well. The reviewers are also satisfied with the writing of the paper, and found the post-rebuttal revision helpful.

On the flip side, the reviewers found the novelty as a weakness, as the central technical components are somewhat standard. But this does not affect the decision of the paper due to TMLR's emphasis on relevance and correctness.

Overall, I recommend the paper to be accepted as is.

**Audience:**

Yes

**Claims And Evidence:**

Yes